# Analysis of particulate emissions from tropical biomass burning using a global aerosol model and long-term surface observations

**C.L.Reddington[1], D.V. Spracklen[1], P. Artaxo[2], D.A. Ridley[1,3], L.V. Rizzo[4] and A. Arana[2]**

[1] {School of Earth and Environment, University of Leeds, Leeds, United Kingdom}

[2] {Department of Applied Physics, Institute of Physics, University of Sao Paulo, Sao Paulo, Brazil}

[3] {now at Department of Civil and Environmental Engineering, Massachusetts Institute of Technology, USA}

[4] {Institute of Environmental, Chemical and Pharmaceutical Sciences, Federal University of Sao Paulo, Diadema, Brazil}

Correspondence to: C. L. Reddington (c.l.s.reddington@leeds.ac.uk)

## Abstract

We use the GLOMAP global aerosol model evaluated against observations of surface particulate matter (PM2.5) and aerosol optical depth (AOD) to better understand the impacts of biomass burning on tropical aerosol over the period 2003 to 2011. Previous studies report a large underestimation of AOD over regions impacted by tropical biomass burning, scaling particulate emissions from fire by up to a factor 6 to enable the models to simulate observed AOD. To explore the uncertainty in emissions we use three satellite-derived fire emission datasets (GFED3, GFAS1 and FINN1) in the model. In these emission datasets the tropics accounts for 66-84% of global particulate emissions from fire. With all emission datasets the model underestimates dry season PM2.5 concentrations in regions of high fire activity in South America and underestimates AOD over South America, Africa and Southeast Asia. When we assume an upper estimate of aerosol hygroscopicity, underestimation of AOD over tropical regions impacted by biomass burning is slightly reduced, relative to previous studies. Where coincident observations of surface PM2.5 and AOD are available we find a greater model underestimation of AOD than PM2.5, even when we assume upper estimates of aerosol hygroscopicity. Increasing particulate emissions to improve simulation of AOD can therefore

lead to overestimation of surface PM2.5 concentrations. With FINN1 emissions scaled by a factor of 1.5 the model reasonably simulates PM2.5 concentrations and AOD in South America and AOD over Southeast Asia, but underestimates AOD over Africa. GFAS1 emissions require a scaling of 3.4 for the model to match observed PM2.5 and AOD, with the exception of Equatorial Asia where a scaling factor of 1.5 is adequate. The model with GFED3 emissions scaled by a factor of 1.5 reasonably simulates PM2.5 concentrations and AOD in active deforestation regions in South America and AOD in Equatorial Asia, but requires a larger scaling factor to capture observed AOD in Africa, Indochina and elsewhere in South America. The model with GFED3 emissions poorly simulates observed seasonal variability of surface PM2.5 and AOD in regions where small fires dominate, providing independent evidence that GFED3 omits emissions from small fires. Seasonal variability of both PM2.5 and AOD in South America is better simulated by the model using FINN1 and GFAS1 emissions. Detailed observations of aerosol properties over biomass burning regions are required to better constrain particulate emissions.

## 1.  Introduction

Open biomass burning is an important source of trace gases and particulate matter (PM) to the atmosphere (Crutzen and Andreae, 1990; Andreae and Merlet, 2001; Van der Werf et al., 2010). Biomass burning emissions can influence weather (Kolusu et al., 2015; Gonçalves et al., 2015; Tosca et al., 2015) and climate (Ramanathan et al., 2001; Tosca et al., 2013; Jacobson, 2014) directly, by scattering and absorbing solar radiation (Johnson et al., 2008; Sakaeda et al., 2011), and indirectly, by modifying cloud properties (Andreae et al., 2004; Feingold et al., 2005; Tosca et al., 2014). The influence of biomass burning aerosol on surface radiation can have subsequent impacts on the biosphere. For example, smoke plumes from biomass burning have been observed to increase plant productivity, through increasing the amount of diffuse radiation (Oliveira et al., 2007; Doughty et al., 2010), which has been shown to be a regionally important process over the Amazon (Rap et al., 2015). PM from biomass burning can substantially degrade regional air quality leading to adverse effects on human health (Emmanuel, 2000; Frankenberg et al., 2005; Johnston et al., 2012; Jacobson, 2014; Reddington et al., 2015).  A better understanding of particulate emissions is needed to improve predictions of the impacts on biomass burning on climate and air quality. Here we use a global aerosol model with tropical observations of surface PM and aerosol optical depth (AOD) to better understand the impact of tropical fires on atmospheric aerosol.

The spatial and temporal distribution of fires depends on climate, vegetation and human activities. At the global scale, fire emissions are dominated by burning in the tropics (van der Werf et al., 2010). Anthropogenic activity can increase the occurrence of fires either directly, through deforestation fires and agricultural residue burning (van der Werf et al., 2010), or indirectly, through land-use/land-cover change that acts to increase the fire susceptibility of the land surface e.g. forest fragmentation in the Amazon (Cochrane and Laurance, 2002) and large-scale drainage of peatlands in Indonesia (Field et al., 2009; Carlson et al., 2012). Human activity can also reduce the occurrence of fires, directly through fire suppression and indirectly through reducing and fragmenting fuel loads which limits fire spread (Bistinas et al., 2014). Over the 21st century, predicted changes in rainfall and temperature may increase forest water stress and subsequent fire occurrence in tropical forests (Cox et al., 2008; Golding and Betts, 2008; Malhi et al., 2009). The incidence of fire and resulting emissions are therefore sensitive both to changing climate and changes in land-use (Heald and Spracklen, 2015).

High temporal and spatial variability in biomass burning emissions coupled with the difficulties involved in conducting measurements in remote tropical regions lead to major challenges for their quantification. In recent years, global estimates of biomass burning emission fluxes have mostly been obtained using satellite remote sensing (e.g., van der Werf et al., 2006, 2010; Reid et al., 2009; Wiedinmyer et al., 2011; Kaiser et al., 2012; Zhang et al., 2012; Ichoku and Ellison, 2014), which provides long-term observations with relatively high spatial coverage. A range of satellite products and methods are utilised to derive fluxes of aerosol and gas-phase species emitted from fires. The most common methods use satellite-retrieved burned area, active fire counts, and/or fire radiative power (FRP) in combination with biogeochemical models (when using burned area) and/or species-specific emission factors obtained from laboratory experiments and field observations (e.g., Hoelzemann et al., 2004; Ito and Penner, 2004; 2005; van der Werf et al., 2006, 2010; Wiedinmyer et al., 2006; 2011; Schultz et al., 2008; Kaiser et al., 2012). Large uncertainties are associated with satellite observations of fires and with the various methods used to calculate emissions fluxes from the observational data (e.g. Ito and Penner, 2005; Reid et al., 2009; Konovalov et al., 2014)

Previous studies using satellite-derived emissions and atmospheric models to investigate the properties and impacts of biomass burning aerosol have found a persistent underestimation of AOD observed in most tropical biomass burning regions (Matichuk et al., 2007; 2008; Chin et al., 2009; Petrenko et al., 2012; Kaiser et al., 2012; Ward et al., 2012; Tosca et al, 2013; Pereira et al., 2016). In general, modelling studies have required biomass burning emissions or

concentrations of biomass burning aerosol to be increased by factors ranging from ~1.5 to ~6 in order to match satellite and ground based observations of AOD (Matichuk et al., 2007; 2008; Johnson et al., 2008; Sakaeda et al., 2011; Johnston et al., 2012; Kaiser et al., 2012; Tosca et al., 2013; Marlier et al., 2013). The underestimation of AOD observed in biomass burning regions has been attributed to a number of factors (see e.g., Kaiser et al., 2012) including: i) underestimation of biomass burning emission fluxes; ii) errors in modelling the atmospheric distribution and properties of biomass burning aerosol; and iii) uncertainties in the calculation of AOD.

Uncertainties associated with the derivation of emission fluxes arise from errors present in the satellite-detection of active fires or burned area (e.g. obscuring of the surface by clouds and smoke, satellite spatial resolution and detection limits, and satellite overpass time), as well as uncertainties in emission factors and fuel consumption estimates. For example, Randerson et al. (2012) suggest that emission datasets based on relatively coarse burned area data (detection limit of ~100 Ha), result in an underestimation of global area burned by ~35%, although this error is not sufficient to fully explain the underestimation of AOD discussed above. Inadequate representation of biomass burning aerosol in models, including errors in the modelled aerosol size distribution, chemical composition, ageing processes, vertical and horizontal transport (including fire emission injection heights) and dry/wet removal from the atmosphere, could also contribute to an underestimation of AOD. The contribution of secondary organic aerosol (SOA) from the oxidation of volatile organic compounds in biomass burning plumes is also a large uncertainty (Jathar et al., 2014; Shrivastava et al., 2015). In the calculation of AOD itself, the uncertainties associated with the assumed optical properties of biomass burning aerosol e.g. their refractive indices, hygroscopicity (uptake of water onto the aerosol), and/or mixing state (i.e. treated as core/shell mixtures, internally/externally mixed etc.) may also contribute to this negative model bias in AOD.

Using only AOD to evaluate estimates of biomass burning aerosol emissions can be misleading because AOD depends on many factors in addition to aerosol abundance. Scaling biomass burning emissions to match observed AOD could therefore lead to inaccurate model representation of biomass burning aerosol concentrations and, subsequently, errors in model predictions of the air quality and climate effects of biomass burning aerosol. Although there has been extensive use of AOD retrievals to evaluate model predictions of biomass burning aerosol, thus far there have been relatively few studies to use aerosol measurements to thoroughly evaluate these models (e.g., Liousse et al., 2010; Daskalakis et al., 2015).

In this study, we evaluate a global aerosol microphysics model against observations of aerosol mass concentrations in addition to AOD to better understand the discrepancy in modelled biomass burning AOD and to ultimately improve estimates of biomass burning aerosol. We also compare three different biomass burning emission inventories to investigate regional differences between emissions and identify the best fit emissions for future modelling studies.

## 2. Observations

To evaluate the simulated distribution of PM at the surface, we use long-term *in-situ* measurements of PM2.5 (particulates with aerodynamic diameters < 2.5 µm) mass concentrations conducted at four ground stations in the Amazon region (Alta Floresta, Porto Velho, Santarem and Manaus). The location and observation period are detailed for each station in Table S1 in the supplementary material. Figure S1 shows the measured PM2.5 concentrations at each station between 2003 and 2011, demonstrating the data coverage.

The PM2.5 measurements were made using gravimetric filter analysis and the measurement duration ranges from less than 1 day to more than 10 days. Particles were sampled under ambient relative humidity (RH) conditions (typically in the range of 80-100% RH). The sampled filters were weighed after 24 hours of equilibration at 50% RH and 20$^o$C. Amazonian submicrometer aerosol particles have growth factors of ~1.1-1.3 at 90% RH (Zhou et al, 2002; Rissler et al., 2006) so we estimate that water represents roughly ~10-20% of the PM2.5 mass concentrations at measurement conditions. Uncertainties related to filter handling, sampling and analysis are estimated as 15% of particle mass. Further information on the measurements conducted at the Manaus and Porto Velho stations can be found in Artaxo et al. (2013). Our evaluation of PM2.5 is restricted to Amazonia since there are few long-term observations of PM2.5 in other tropical regions impacted by biomass burning.

The measurement stations at Porto Velho and Alta Floresta are located in the arc of deforestation and are strongly impacted by fresh biomass burning emissions (Fig. 1). The Santarem and Manaus stations are located within forest reservations and are impacted by transported regional biomass burning emissions in the dry season. The Santarem station is located in Para, where the number of fire hotspots observed by satellites during the dry season are typically a factor of ~10 great than the number observed in Amazonas, where the Manaus station is located. Thus in the dry season, PM2.5 concentrations measured at Santarem are typically higher than those measured at Manaus.

To evaluate the simulated distribution of AOD, we use observations of spectral columnar AOD measured by the Aerosol Robotic Network (AERONET) using ground-based Cimel sun photometers (Holben et al., 1998). Specifically, we use Level 2.0 (quality assured) daily average AOD retrieved at 440 nm from 27 AERONET stations detailed in Table S1. We selected stations located within regions influenced by tropical biomass burning (Southeast and Equatorial Asia, Central and Southern Africa, and the Amazon region in South America) that have more than one year of relatively continuous data (automatic cloud screening leads to gaps in the dataset) between 2003 and 2011. We note that whilst the majority of cloud-contaminated AOD data is removed; comparisons with co-located Micro-Pulse Lidar Network observations indicate that some contamination from thin cirrus clouds may remain, possibly leading to small positive biases in observed AOD (Huang et al., 2011; Chew et al., 2011).

To compare modelled and observed PM2.5 and AOD, daily-mean model output was linearly interpolated to the location (latitude, longitude and altitude above sea level) of each ground station. Model data that corresponded to gaps in the observation datasets were removed prior to calculating monthly-mean values used in the analysis. The modelled PM2.5 concentration is calculated for dry aerosol, omitting the contribution of water to the total mass, thus modelled PM2.5 concentrations may be underestimated compared to the observations, which include some contribution from the mass of water.

## 3. Model description

### 3.1 Global aerosol microphysics model

The global distribution of aerosol was simulated using the 3-D Global Model of Aerosol Processes (GLOMAP; Spracklen et al., 2005a,b; Mann et al., 2010), which is an extension to the TOMCAT chemical transport model (Chipperfield, 2006). Simulations were run for the period 2003 to 2011. Large scale atmospheric transport and meteorology in TOMCAT are specified from European Centre for Medium-Range Weather Forecasts (ECMWF) analyses, updated every 6 hours and linearly interpolated onto the model time-step. The model runs at a horizontal resolution of 2.8°×2.8° with 31 vertical model levels between the surface and 10 hPa. The vertical resolution in the boundary layer ranges from ~60 m near the surface to ~400 m at ~2 km above the surface. GLOMAP has been extensively evaluated in previous studies against aerosol observations (Mann et al., 2010, 2014; Spracklen et al., 2011a,b; Schmidt et al., 2012; Scott et al., 2014; Reddington et al., 2011, 2013, 2014). Below we describe the features of the

model relevant for this study, please see Spracklen et al. (2005a) and Mann et al. (2010) for more detailed descriptions of the model.

GLOMAP simulates the mass and number of size resolved aerosol particles in the atmosphere, including the influence of aerosol microphysical processes on the particle size distribution. These processes include nucleation, coagulation, condensation, ageing, hygroscopic growth, cloud processing, dry deposition, and nucleation/impact scavenging. The aerosol particle size distribution is represented using a two-moment modal scheme with seven log-normal modes (Mann et al., 2010). Within each mode, aerosol particles are treated as internally mixed. GLOMAP treats the following aerosol species: black carbon (BC), particulate organic matter (POM), sulphate ($SO_4$), sea spray and mineral dust. Biogenic SOA is formed in the model via the reaction of biogenic monoterpenes with $O_3$, OH and $NO_3$, which produces a gas-phase oxidation product that condenses with zero vapour pressure onto pre-existing aerosol (Spracklen et al., 2006, 2008). Concentrations of oxidants are specified using monthly-mean 3-D fields at 6-hourly intervals from a TOMCAT simulation with detailed tropospheric chemistry (Arnold et al., 2005) linearly interpolated onto the model time-step. Monthly mean emissions of biogenic monoterpenes are taken from the Global Emissions InitiAtive (GEIA) database (Guenther et al., 1995). Size-resolved emissions of mineral dust are prescribed from daily-varying emissions fluxes provided for AEROCOM (Dentener et al., 2006).

For this study, anthropogenic emissions of sulphur dioxide ($SO_2$), BC and organic carbon (OC) were specified using the MACCity emissions inventory (Lamarque et al., 2010; Granier et al., 2011), which provides annually varying emissions for the period 1979-2010. For simulations in the year 2011 we used MACCity anthropogenic emissions from 2010. Biomass burning emissions of $SO_2$, BC and OC were specified using three different satellite-derived emission datasets, which are described in detail in Section 3.3. We convert OC to POM using a prescribed POM:OC ratio of 1.4, which is at the lower end of the range prescribed in other global models (1.4 to 2.6) (Tsigaridis et al., 2014). The fire emissions were injected into the model over six ecosystem-dependent altitudes between the surface and 6 km recommended by Dentener et al. (2006). In the regions studied in this paper (South America, Africa and Southeast Asia), the fire emission injection heights range between the surface and an altitude of ~3 km asl. The largest fraction of the fire emissions, ranging from ~99% of emissions in Equatorial Asia to 88% in Indochina, are injected below 1 km asl (or at surface level if the altitude of the model level exceeds 1 km asl). Analysis of smoke plume heights has demonstrated that most smoke emissions from fires occur within the boundary layer (Val Martin et al., 2010).

Primary carbonaceous aerosol particles are assumed to be non-volatile and are emitted into the model with a fixed log-normal size distribution, assuming a number median diameter of 150 nm for biomass burning emissions and 60 nm for fossil fuel emissions and modal width (σ) of 1.59. Several previous studies have investigated the impacts of the uncertainty in the assumed emission size distribution on simulated aerosol and cloud condensation nuclei concentrations (Pierce et al., 2007; Pierce and Adams, 2009; Reddington et al., 2011; 2013; Lee et al., 2013) and aerosol radiative forcing (Bauer et al., 2010; Spracklen et al., 2011b; Carslaw et al., 2013). An assumption of a number median diameter of 150 nm for biomass burning emissions is reasonably consistent with measurements of the size distributions of fresh biomass burning aerosol from grassland (100 – 125 nm) and deforestation (100 – 130 nm) fires (Reid et al., 2005 and references therein). Once emitted into the model, the components of primary carbonaceous aerosol (BC and OC) are assumed to mix instantaneously and are initially treated as non-hygroscopic. Once these particles have accumulated 10 monolayers of soluble material (assumed to be SOA and $H_2SO_4$) through condensation, they are transferred directly to the corresponding soluble Aitken or accumulation mode to account for ageing. For a discussion of the treatment of organic aerosol within global aerosol models see Tsigaridis et al. (2014).

## 3.2   Calculation of aerosol optical depth

AOD was calculated from the simulated aerosol size distribution using Mie theory assuming spherical particles (Grainger et al., 2004) that are externally mixed within each log-normal mode. For this study, modelled AOD was calculated at a wavelength of 440 nm using component-specific refractive indices at the closest wavelength available (468 nm) from Bellouin et al. (2011). Water uptake plays a significant role in determining AOD, altering the refractive index and the size distribution of the aerosol. The water uptake for each soluble aerosol component is calculated on-line in the model according to Zdanovskii-Stokes-Robinson (ZSR) theory, which estimates the liquid water content as a function of solute molarity (Stokes and Robinson, 1966). For POM in the soluble modes, we assign a hygroscopicity consistent with a water uptake per mole at 65% of that of $SO_4$ (Mann et al., 2010). This is an upper estimate of aerosol hygroscopicity. In section 4.1.3 we explore the sensitivity of simulated AOD to different assumptions on aerosol hygroscopicity as well as aerosol refractive indices and aerosol mixing state. The resulting daily-mean wet radii and refractive indices are used to calculate the daily-mean aerosol extinction. Using hourly-mean values of water uptake increased simulated daily AOD on average by less than 1%.

## 3.3 Biomass burning emissions

In this study we compare three different satellite-derived datasets of biomass burning emissions: the Global Fire Emissions Database version 3 (GFED3; van der Werf et al., 2010), the National Centre for Atmospheric Research Fire Inventory version 1.0 (FINN1; Wiedinmyer et al., 2011) and the Global Fire Assimilation System version 1.0 (GFAS1; Kaiser et al., 2012). The key aspects of these emission inventories are summarised in Table 1. We complete GLOMAP simulations for the period 2003 to 2011 where all three emission datasets are available.

GFED3 provides monthly-mean fire emissions of aerosol and gas-phase species from 1997 to 2011 at 0.5°×0.5° resolution (van der Werf et al., 2010). GFED3 emissions are derived using the monthly-mean time series of global burned area estimates from Giglio et al. (2010). For 1997-2000, the fire emissions are based on burned area derived from the TRMM Visible and Infrared Scanner (VIRS) and Along-Track Scanning Radiometer (ATSR) active fire data and estimates of plant productivity derived from observations from the Advanced Very High Resolution Radiometer (AVHRR). For November 2000 onwards, the fire emissions are based on estimates of burned area, active fire detections, and plant productivity from the MODerate resolution Imaging Spectroradiometer (MODIS) instrument on-board the Terra and Aqua satellites. To derive total carbon emissions the satellite datasets are combined with estimates of fuel loads and combustion completeness for each monthly time step from the Carnegie-Ames-Stanford-Approach biogeochemical model. The carbon emission fluxes are converted to trace gas and aerosol emissions using species specific emission factors complied by Andreae and Merlet (2001). From 2003 onwards, GFED3 fire emissions are available on a daily time step, developed using detections of active fires from MODIS (Mu et al., 2011). Daily GFED3 fire emissions were implemented in GLOMAP for the period 2003-2011.

FINN1 provides daily fire emissions of aerosol and gas-phase species from 2002 to 2012 on a 1 km$^2$ grid (Wiedinmyer et al., 2011). FINN1 fire emissions are based on detections of active fires (specifically their location and timing) from the MODIS Fire and Thermal Anomalies Product (Giglio et al., 2003). FINN1 also uses the MODIS Land Cover Type product to specify land cover classes and the MODIS Vegetation Continuous Fields product to identify the fractions of tree and non-tree vegetation, and bare ground. Specifically, the emitted mass (*E*) of a certain species (*i*) is calculated using the following equation (Seiler and Crutzen, 1980):

$$E_i = A(x,t) \times B(x) \times FB \times ef_i \qquad\qquad (1)$$

Where $A$ is the area burned at time $t$ and location $x$, $B$ is the biomass loading at location $x$, $FB$ is the fraction of that biomass that is burned and $ef$ is the emission factor of species $i$. For each fire count the area burned, $A$, is assumed to be 0.75 km$^2$ for fires detected on grassland and savannah land cover classes, and 1 km$^2$ for those detected on all other land cover classes following Wiedinmyer et al. (2006) and Al-Saadi et al. (2008). Adjustments are made to the assumed burned area if the fire pixel extends partially over bare ground (reducing the burned area by the percentage of bare area assigned to that pixel). Estimates of biomass loading, $B$, are taken from Hoelzemann et al. (2004) and are assumed to be land cover specific. The fraction of biomass assumed to burn, $FB$, in each fire pixel is determined as a function of tree cover using relationships from Ito and Penner (2004) (see Wiedinmyer et al., 2006). Emission factors, $ef$, for each species are taken from Akagi et al. (2011).

GFAS1 provides daily fire emissions of aerosol and gas-phase species from March 2000 to 2013 at 0.5°×0.5° resolution (Kaiser et al., 2012). Like FINN1, GFAS1 uses the observed geo-location of active fires from the MODIS instrument. However, GFAS1 also makes use of the NASA fire products (MOD14 and MYD14) that provide quantitative information on the radiative power of detected fires (Justice et al., 2002; Giglio, 2005). The FRP fields are corrected for observation gaps due to partial cloud-cover by assuming the same FRP areal density throughout the grid cell. Data assimilation is used to further fill observation gaps using information from earlier FRP observations (see Kaiser et al., 2012). Spurious signals from volcanoes, gas flares and other industrial activity are removed from the data. The FRP is converted to the combustion rate of dry matter using land-cover-specific conversion factors based on data from GFED3 (Heil et al., 2010; Kaiser et al., 2012). As for GFED3, species emission rates are calculated using updated emission factors based on Andreae and Merlet (2001).

Table 1 gives the total annual amounts of BC and OC aerosol emitted from fires over the tropics for each emission inventory. The total BC and OC emitted from fires in the tropics make up 77-84% and 66-77%, respectively of the global total emissions. FINN1 has the greatest tropical OC emission, with emissions being 47% greater than in GFAS1 and 30% greater than GFED3. Emission of BC is more consistent, with FINN1 BC emissions being 13% greater than GFAS1 and 1% greater than GFED3. This results in different OC:BC emission ratios between the datasets with the mean ratio across the tropics varying from 10.0 in FINN1, 7.9 in GFED3 and 7.1 in GFAS1.

Figure 1a-c shows the spatial distribution of annual total biomass burning emissions of OC from each fire inventory averaged over the period of 2003 to 2011. There are similarities in the general spatial distributions of fire emissions, with all three inventories showing maximum emissions over the tropical savannah and humid subtropical regions of Africa, the arc of deforestation in Amazonia, coastal regions of Indonesia (Sumatra and Kalimantan), northern Australia, and parts of Indochina (particularly Cambodia, Laos and Myanmar). However, Figs. 1d-f show that there are strong regional differences between the different emission inventories. Differences between FINN1 and GFAS1 (Fig. 1e) and FINN1 and GFED3 (Fig. 1f) are more spatially organised than differences between GFAS1 and GFED3 (Fig. 1d), which are more spatially heterogeneous.

Over Africa, GFED3 gives higher OC emissions in northern tropical savannah and southern humid subtropical regions, with GFAS1 and FINN1 giving higher emissions than GFED3 at the boundaries of these regions and over central Africa. Over Australia, GFED3 gives the highest OC emissions estimates over the tropical savannah region of northern Australia, with GFAS1 giving the highest emissions in the dryer grassland and desert regions further south.

Over South America the picture is more complex. In general, FINN1 and GFAS1 emission estimates are higher in northern and eastern Brazil than GFED3, with GFAS1 giving the highest emissions over eastern areas and FINN1 over northern Brazil. FINN1 emissions are generally higher than GFAS1 and GFED3 over the central and southern Amazon region (particularly over the state of Mato Grosso), Peru and generally over northern South America. GFED3 emissions are higher than FINN1 and GFAS1 in northern parts of Bolivia and the northern part of the state of Rondônia in the arc of deforestation.

Over South Asia, Indochina and Equatorial Asia, FINN1 gives higher emissions than both GFED3 and GFAS, particularly over Bangladesh, Myanmar and Laos, with the exception of the coastal peatland regions of Sumatra and Kalimantan where GFAS1 and GFED3 give higher emissions than FINN1. The differences in emissions over Indonesia may be explained by a potentially improved representation of tropical peat fire emissions in GFED3 and GFAS1 relative to FINN1 (Andela et al., 2013).

**4. Results**

**4.1 Overview of all comparisons**

**4.1.1 Particulate matter concentrations in the Amazon region**

Figure 2 shows simulated versus observed multi-annual monthly mean PM2.5 concentrations at each of the four ground stations in the Amazon region (see Fig. 1 for site locations). To quantify the agreement between model and observations, we use the Pearson correlation coefficient (r) and normalised mean bias factor (NMBF) as defined by Yu et al. (2006):

$$NMBF = \frac{(\sum M_i - \sum O_i)}{|\sum M_i - \sum O_i|} \left[ \exp\left( \left| \ln \frac{\sum M_i}{\sum O_i} \right| \right) - 1 \right]$$

where $M$ and $O$ represent the multi-annual monthly mean model and observed values, respectively, for each month $i$. A positive NMBF indicates the model overestimates the observations by a factor of NMBF+1. A negative NMBF indicates the model underestimates the observations by a factor of 1–NMBF.

Figure 2 demonstrates the important contribution of biomass burning to PM2.5 concentrations across the region: there is a strong improvement in the agreement between model and observations when biomass burning emissions are included in the model (Fig. 2b-d; NMBF =- 0.62 to -0.25, $r^2$=0.77-0.83) relative to the simulation without fire emissions (Fig. 2a; NMBF= -1.85, $r^2$=0.44).

The overall bias between model and observations is smallest with FINN1 emissions (NMBF= -0.25) compared to GFED3 (NMBF= -0.49) or GFAS1 (NMBF= -0.62), with simulated monthly mean concentrations mostly within a factor of ~2 of the observations. The correlation between model and observations across all sites is relatively similar between the three emission datasets, with a slightly stronger correlation with GFED3 emissions ($r^2$=0.83) compared to FINN1 ($r^2$=0.77) and GFAS1 ($r^2$=0.79).

The NMBF and correlation between model and observations are shown for the individual stations in Fig. 3a. Correlations are calculated between simulated and observed multi-annual monthly mean concentrations to evaluate the ability of the model to simulate seasonal variability in aerosol. In general, the model with fire emissions overestimates observed PM2.5 concentrations at the forest site near Manaus (mean NMBF=0.57) but underestimates observed PM2.5 concentrations at the sites that are more strongly impacted by biomass burning (Porto

Velho, Alta Floresta and Santarem; mean NMBF= -0.60). Figure 3 demonstrates that the relatively small bias with the FINN1 emissions in Fig. 2 is partly due to an overestimation of PM2.5 concentrations at Manaus (NMBF=0.98), but also due to smaller model biases at the three other sites (-0.51 to -0.11) compared to GFED3 (-0.76 to -0.48) and GFAS1 (-1.26 to -0.39).

Figure 4 shows the multi-annual average seasonal cycle in observed and simulated PM2.5 concentrations at the four measurement sites (the full time-series at each site is shown in Fig. S1 in the supplementary material). The model with biomass burning emissions simulates the observed seasonal variability in PM2.5 concentrations over the Amazon region, characterised by high concentrations in the local dry season (between ~June to ~December depending on the site) and relatively low concentrations in the wet season. At Porto Velho, Santarem and Alta Floresta, the model underestimates observed PM2.5 concentrations during the dry season and has relatively good agreement during the wet season. This suggests that the negative model bias in the dry season is largely due to uncertainty in the biomass burning emissions rather than anthropogenic emissions, biogenic SOA or microphysical processes in the model. The model overestimates PM2.5 concentrations observed at Manaus all year round, but particularly during the dry season. This positive model bias may be due to several factors including a possible overestimation of biogenic SOA over tropical forests and/or the model resolution, which is not fully capturing the gradient in PM2.5 concentrations between the arc of deforestation and the relatively undisturbed forest near Manaus.

In previous work we carried out a detailed model sensitivity analysis that accounted for the uncertainty in the emissions (including biomass burning) and in the model processes such as wet removal and dry deposition of aerosol (Lee et al., 2013). This analysis confirms that the parametric uncertainty in modelled PM2.5 concentrations at these four stations is dominated by the uncertainty in the biomass burning emissions flux in the dry season and by the yield of biogenic SOA in the wet season, rather than the removal processes in the model.

Figure 4 demonstrates the differences in the spatial and temporal variability between the three fire emission datasets, with different emissions capturing the observations better in different months and locations. The model with GFED3 emissions captures the average seasonal variability in PM2.5 observed at Alta Floresta (Fig. 4; $r^2$=0.69) and Porto Velho ($r^2$=0.94) reasonably well. In particular, better simulating the peak in dry season concentrations at Porto Velho than both FINN1 ($r^2$=0.72) and GFAS1 ($r^2$=0.85) emissions. However, PM2.5 concentrations observed towards the end of the biomass burning season at Alta Floresta

(September – November) and Porto Velho (October – November) are not well captured by GFED3 emissions, leading to larger biases at these sites (NMBF= -0.73 and -0.48, respectively) than with FINN1 emissions (-0.51 and -0.41, respectively). At Santarem, the model with GFED3 emissions underestimates observed PM2.5 concentrations throughout the dry season, leading to a relatively large model bias and poor correlation with the observations (NMBF= -0.76, $r^2$=0.39). Agreement with the observations at this site is improved with either FINN1 (NMBF= -0.11, $r^2$= 0.76) or GFAS1 (NMBF= -0.39, $r^2$= 0.75) emissions (discussed further in Sect. 4.2).

If we consider the inter-annual variability in simulated and observed PM2.5 concentrations (Figure S2), we find that the results are consistent with the evaluation of the simulated seasonal cycle. The smallest bias between model and observations is with the FINN1 emissions (NMBF= -0.22) compared to GFED3 (NMBF= -0.36) or GFAS1 (NMBF= -0.48). One notable point is that the model with GFED3 emissions simulates the highest PM2.5 concentrations for the 2010 drought year, relative to the model with GFAS1 or FINN1 emissions, leading to improved agreement with observations at Porto Velho (see Figs. 3a, 4a and S2).

In summary, the model captures the seasonal cycle and inter-annual variability of observed PM2.5 reasonably well at biomass burning influenced sites in the Amazon. However, the model underestimates observed concentrations in the dry season suggesting that the biomass burning aerosol emission fluxes in all three emission inventories (GFED3, FINN1, GFAS1) may be underestimated. We explore this further in Section 4.3.

### 4.1.2  Aerosol optical depth in tropical biomass burning regions

Figure 5 shows the simulated versus observed multi-annual monthly mean AOD at 440 nm at each of the AERONET sites displayed in Fig. 1 (simulated and observed annual means are compared in Fig. S3). Agreement between model and observed AOD is improved substantially when biomass burning emissions are included in the model (Fig 5; NMBF= -0.40 to -0.18, $r^2$=0.62-0.69) compared to the simulation without fire emissions (NMBF= -0.69, $r^2$=0.22). As for PM2.5, the bias in AOD across all sites is smallest with the FINN1 emissions (NMBF= -0.18) compared to GFED3 (NMBF= -0.34) or GFAS1 (NMBF= -0.40). The model with FINN1 emissions also shows slightly improved correlation with the observations ($r^2$=0.69) relative to GFED3 ($r^2$=0.67) and GFAS1 ($r^2$=0.62).

Figure 6a shows the NMBF and correlation between simulated and observed multi-annual monthly mean AOD at the individual AERONET sites, grouped by region. In South America,

the bias in modelled AOD is smallest with the FINN1 emissions (mean NMBF= -0.47) compared to GFED3 (-0.69) and GFAS1 (-0.89) emissions, which is consistent with comparisons between modelled and observed PM2.5 in Amazonia (Sect. 4.1.1). In Indochina, the model with FINN1 emissions also gives the smallest bias (mean NMBF= -0.02), relative to GFED3 (-0.21) and GFAS1 (-0.23). In Africa, the model bias is smallest with GFED3 emissions (mean NMBF= -0.78) compared to GFAS1 (-0.90) and FINN1 (-0.96). In Equatorial Asia, the model bias is small and does not vary substantially between the different emission datasets (FINN: 0.02, GFAS: -0.01, GFED: -0.02). In terms of temporal agreement between model and observations, the correlation is noticeably stronger with GFED3 (mean $r^2$ =0.52) in Africa and with FINN1 (mean $r^2$=0.75) in Indochina, relative to the other emission datasets.

In general, the model with fire emissions captures the seasonal variability in observed AOD best in South America (mean $r^2$=0.90) and captures the magnitude of observed AOD best in Southeast Asia (Equatorial Asia: mean NMBF= -0.00; Indochina: mean NMBF= -0.14). The agreement between model and observations in Africa is relatively poor, with substantial underestimation of observed AOD (mean NMBF= -0.88). The negative model bias in Africa is unlikely to be solely due to an underestimation of biomass burning aerosol and is likely complicated by a contribution from dust (Pandithurai et al., 2001; Sayer et al., 2014; Cesnulyte et al., 2014; Queface et al., 2011). There is better agreement between the model and observed AOD at Ascension Island, which observes aged biomass burning aerosol from the African continent (Sayer et al., 2014), with all three emission inventories (mean NMBF= -0.38, $r^2$=0.84). This suggests that the model is able to capture outflow of biomass burning emissions from Africa.

At the South American sites located in regions of high biomass burning activity associated with deforestation fires (Abracos Hill, Rio Branco, Ji Parana SE and Alta Floresta), there is a small improvement in the correlation with observed AOD with FINN1 ($r^2$=0.96-0.98) and GFAS1 ($r^2$=0.94-0.97) emissions relative to GFED3 ($r^2$=0.79-0.88). At these sites, AOD observed at the tail end of the biomass burning season (~October-November) is better captured by GFAS1 and FINN1 than GFED3, leading to the improved correlation relative to GFED3. The model with GFED3 is generally better able to capture observed AOD at the peak of the biomass burning season (~August-September) than GFAS1 and FINN, which is largely due to relatively high GFED3 emission estimates for the drought years 2007 and 2010 (see Fig. S1). These results are consistent with comparisons with observed PM2.5 concentrations at Porto Velho and Alta Floresta (Sect. 4.1.1).

At the AERONET sites located in Equatorial Asia and the Philippines (Singapore, Bandung, Manila Observatory, ND Marbel Univ) an improved performance of either the GFAS1 or GFED3 emission inventories may be expected over FINN1 (Andela et al., 2013) due to their improved representation of tropical peatlands (in Indonesia and Malaysian Borneo) in their biome maps (van der Werf et al., 2010). The agreement between AOD observed at Bandung, Indonesia and the model is marginally improved with GFED3 (NMBF= -0.14, $r^2$=0.52) or GFAS1 (NMBF= -0.15, $r^2$=0.47) relative to FINN1 (NMBF= -0.18, $r^2$=0.34). However, at the other sites we find no strong indication of an improved performance with GFED3 (NMBF= -0.06 to 0.13, $r^2$=0.15-0.24) or GFAS1 (NMBF= -0.03 to 0.14, $r^2$=0.13-0.56) relative to FINN1 (NMBF= 0.04 to 0.17, $r^2$=0.16-0.42). At most of these sites the model does not simulate a strong contribution of biomass burning to AOD, likely due to their urban locations, which may explain why we do not see a substantial difference in the performances of the three emission datasets. Long-term ground-based retrievals of AOD located outside the influence of urban environments are lacking in Equatorial Asia.

At the African AERONET sites, observed AODs are generally better captured by the model with GFED3 emissions (mean NMBF= -0.78, $r^2$=0.52) than with FINN1 (mean NMBF= -0.96, $r^2$=0.35) or GFAS1 (mean NMBF= -0.90, $r^2$=0.41) emissions. Andela et al. (2013) report that the GFED3 emissions flux of carbon monoxide (CO) is higher than GFAS1 or FINN1 for humid savannah regions, where the burned area product may observe more cloud covered fires than active-fire detection. This feature may explain the improved simulation of AOD with GFED3 over Africa. Andela et al. (2013) also report that the FINN1 emission estimates of CO are lower than both GFED3 and GFAS1 in global savannah regions, with the largest spatial deviation found in humid savannahs where fire size is large. This may suggest that the assumed fire size in FINN1 for savannah fires (0.75 km$^2$) could be too small for humid savannah fires in Africa, contributing to an underestimation of AOD in this region.

### 4.1.3  Overview of PM2.5 and AOD evaluation

In the previous sections we have evaluated the model against ground based observations of PM2.5 and AOD. In general, we find that the model is negatively biased against observations in regions strongly influenced by biomass burning. However, the model bias in surface PM2.5 concentrations is generally smaller than for AOD over South America, where observations of both quantities are available (NMBF$_{PM2.5}$= -1.85 to -0.25, NMBF$_{AOD}$= -2.38 to -0.40; see Figs. 2 and S4). If we compare average model biases (with fires) in multi-annual monthly mean

PM2.5 and AOD (for 2003-2004) at locations where AERONET stations are in close proximity to the PM2.5 measurement stations, we find a larger model bias in AOD at Santarem/Belterra ($NMBF_{PM2.5}$ = -0.61, $NMBF_{AOD}$ = -1.15), but the reverse at Alta Floresta ($NMBF_{PM2.5}$ = -0.64, $NMBF_{AOD}$ = -0.42).

These results suggest that although the negative model bias in PM2.5 and AOD may be partly due to an underestimation of biomass burning aerosol emissions (due to uncertainties associated with fire detection and subsequent calculations of emission fluxes), there are likely to be other factors contributing to the model discrepancy in AOD that do not affect modelled surface PM2.5 concentrations. These factors include uncertainties in the calculation of AOD that are largely associated with assumptions made about the aerosol optical properties (assumed refractive indices), mixing state (external/internal mixing) and hygroscopic growth of the aerosol. We investigate the sensitivity of simulated AOD to these assumptions below.

As described in Sect. 3.2, to calculate AOD at 440 nm we use component-specific refractive indices from Bellouin et al. (2011) for a wavelength of 468 nm ($1.500 - 0.000i$ for POM and $1.750 - 0.452i$ for BC). To test the sensitivity of AOD to the choice of refractive indices, we applied the refractive indices tested by Matichuk et al. (2007) for smoke aerosol ($1.54 - 0.025i$ calculated by Haywood et al. (2003) for young smoke aerosol over southern Africa; $1.51 - 0.024i$ and $1.52 - 0.019i$ retrieved by an AERONET station, Ndola in Zambia, located close to smoke sources) to the BC and POM components in our model., We find that the modelled AOD is relatively insensitive to the choice of complex refractive index within the range of values tested here (altering the magnitude of AOD by less than 5%), which is in agreement with Matichuk et al. (2007). Although the range of refractive indices tested is relatively narrow (Matichuk et al., 2007), this result suggests that uncertainty in the assumed refractive indices is unlikely to explain the discrepancy in simulated AOD.

We also find that the AOD is fairly insensitive to the mixing state assumption, with limited difference in simulated AOD between assuming optical properties derived from an external mixture of aerosol species and an internal (volumetrically-averaged) mixture. Figure S5 shows the simulated versus observed multi-annual monthly mean AOD at AERONET sites when assuming external and internal mixing and indicates that the difference is less than 5%, with internal mixing causing slightly higher AOD at the AERONET sites. However, we note that the internal mixing assumption used in this study does not take into account the lensing effects of coating BC with organic aerosol, which has been shown to interact with the aerosol absorption in a non-linear way (Saleh et al., 2015).

As described in Sect. 3.2, the hygroscopic growth of the aerosol is calculated in the model using the ZSR scheme. To test the sensitivity of AOD to aerosol hygroscopic growth, we instead use the κ-Köhler water uptake scheme, based upon the Köhler equation with a single hygroscopic parameter, κ, defining the water uptake for different chemical species (Petters and Kreidenweis, 2007) (see description of method in Sect. S1 of the supplementary material). For the $SO_4$ and sea spray components in the model we used the mean values of κ for ammonium sulphate and sodium chloride for subsaturated air masses (0.53 and 1.12, respectively) from Petters and Kreidenweis (2007). BC is considered entirely hydrophobic in this model when using this scheme. A wide range of κ values have been reported for organic aerosol (~0.01-0.25; Petters and Kreidenweis, 2007) and biomass burning particles specifically (0.02-0.8; DeMott et al., 2009; Petters et al., 2009). Engelhart et al. (2012) reported κ values of between 0.06 and 0.6 for primary biomass burning aerosol in a smog chamber (fuels representative of North American wildfires), with photochemical ageing reducing the range of κ values to 0.08 to 0.3, with biomass burning SOA having κ values of 0.11. We assume a κ value for POM (0.1) based upon aerosol samples, largely composed of SOA, collected at the Manaus ground station (TT34) during the 2008 Amazonian Aerosol Characterization Experiment (AMAZE-08) (Gunthe et al., 2009). We test the sensitivity of simulated AOD to different κ values for both $SO_4$ and POM.

Figure 7 shows a comparison between AOD simulated using ZSR and the κ-Köhler scheme. Using the κ-Köhler scheme and κ defined above, the water uptake is reduced relative to the ZSR scheme, reducing the simulated AOD on average by a factor of 1.6 (range 1.1 to 2.3) at AERONET sites (see Figs. 7a and 7b). This large reduction relative to ZSR is in part from the assumption that the $SO_4^{2-}$ component behaves as ammonium sulphate rather than the more hygroscopic sulphuric acid, and the reduced water uptake for POM. To explore the sensitivity to assumed κ values we increased κ values separately for $SO_4$ and POM. Assuming a higher κ for sulfate (1.19 as for sulphuric acid, Fig, 7c) results in simulated AOD being a factor 1.25 lower than ZSR. Assuming a higher κ for both sulfate (1.19) and for POM (0.2) results in simulated AOD being a factor of 1.18 lower. Our results highlight the large uncertainty present in the simulated AOD due to aerosol hygroscopicity. AOD simulated with ZSR (assuming sulfuric acid and high water uptake for organics) appears to be an upper estimate for water uptake. This result is confirmed by comparing simulated AOD and mass extinction efficiencies for the two water uptake cases against observations and values from other global aerosol models (see Sect. S2 and Table S2).

Calculated AOD is also sensitive to errors in relative humidity (Myhre et al., 2009), which are here taken from ECMWF re-analysis. Since water uptake is not a linear function of RH, calculated AOD will also be sensitive to spatial resolution of the aerosol and RH fields. Coarse spatial resolution (here 2.8°) will not capture fine scale variability in RH that will influence measurements from AERONET stations. A higher resolution model would be required to test how sensitive the simulated AOD is to the spatial resolution of the aerosol and RH fields and whether or not increasing the resolution improves the agreement with observed AOD (and reduces the discrepancy between the model performance in AOD and PM2.5). Bian et al. (2009) showed that increasing the resolution of the RH field from 2°x2.5° to 1°x1.25° can increase simulated AOD by ~10% in biomass burning regions. This suggests the coarse resolution of our global models may partly explain the underestimation of AOD and the larger discrepancies with observed AOD compared to PM2.5.

Errors may also exist in the model representation of biomass burning aerosol, for example in the modelled particle size distribution, altering simulated optical properties of the aerosol and thus calculated AOD. In addition, since AOD is a column-integrated quantity, an underestimation of AOD may be due to an underestimation of aerosol concentrations aloft since we have shown that the model agrees relatively well with PM2.5 concentrations observed at the surface.

Further uncertainties in the model representation of biomass burning aerosol are associated with the conversion of OC to organic matter (OM), which would affect both PM2.5 concentrations and AOD predicted by the model. Increasing the assumed OM:OC ratio would increase the total simulated mass of biomass burning aerosol. In our model we assume a relatively low OM:OC ratio of 1.4 compared to previous studies on biomass burning aerosol. Kaiser et al. (2012) use a value of 1.5, but note this ratio is low compared to values of around 2.2 proposed for aged pollution and biomass burning aerosols by Turpin and Lim (2001), Pang et al. (2006) and Chen and Yu (2007) and a value of 2.6 used by Myhre et al. (2003) for biomass burning aerosol in southern Africa. These larger OM:OC ratios could account for in-plume (sub-grid) atmospheric oxidation and subsequent SOA formation observed in some biomass burning plumes (Vakkari et al., 2014). In future work we need to include the formation of semi-volatile SOA in biomass burning plumes that has been shown to be important (Konovalov et al., 2015; Shrivastava et al., 2015).

## 4.2 Small-scale fires

The GFED3 fire emissions are known to underestimate contributions from small-scale fires (smaller than ~100 ha) that are below the detection limit of the global burned area product derived from MODIS (Randerson et al., 2012). However, many of these small fires generate thermal anomalies that can be detected by satellites (Randerson et al., 2012). This means that fire inventories using active fire detections to derive emissions (FINN1 and GFAS1) will better capture these small fires (Kaiser et al., 2012). Kaiser et al. (2012) demonstrate that GFAS1 includes emissions from small fires that are omitted in GFED3. Some of the differences between the spatial patterns of emissions seen in Fig. 1 are likely due to missing small fires in GFED3.

This result is corroborated by our comparisons between modelled and observed PM2.5 concentrations at Santarem in the north region of Brazil (Sect. 4.1.1), where the poor agreement between the observations and model with GFED3 emissions (NMBF= -0.76, $r^2$=0.39) is substantially improved by using either of the active-fire based emission inventories (FINN: NMBF= -0.11, $r^2$= 0.76; or GFAS: NMBF= -0.39, $r^2$= 0.75). Randerson et al. (2012) show that in the region surrounding the Santarem station there is a particularly high small fire fraction of total burned area, which explains why the GFED3 emissions do not capture the observations in this region of Brazil. This result is consistent with comparisons between modelled and observed AOD at the nearby AERONET station, Belterra. At this station, the model better captures the observed AOD with either FINN1 (NMBF= -0.85, $r^2$=0.84) or GFAS1 (NMBF= -1.02, $r^2$=0.81) emissions than with GFED3 emissions (NMBF= -1.58, $r^2$=0.29).

The improved representation of small fire emissions in FINN1 and GFAS1 may also explain the improved agreement between modelled and observed PM2.5 (Sect. 4.1.1) and AOD (Sect. 4.1.2) towards the end of the burning season (~October-November) in Amazonia. Kaiser et al. (2012) report that GFAS1 exhibits slightly longer fire seasons in South America than GFED3. Fires occurring at the tail end of the biomass burning season may be smaller in size and thus better captured by using an active-fire based emission inventory (GFAS1 and FINN1 emissions). While at the peak of the burning season in Amazonia, when fires are potentially larger, the comparisons in Sects. 4.1.1 and 4.1.2 suggest that GFED3 emissions capture the observations better than FINN1 or GFAS1.

In Indochina, there is improved agreement between simulated and observed AOD with FINN1 emissions (Fig. 6a; NMBF= -0.26 to 0.19, $r^2$=0.14-0.98) relative to both GFED3 (NMBF= -

0.54 to -0.08, $r^2$=0.11-0.84) and GFAS1 (NMBF= -0.51 to -0.08, $r^2$=0.03-0.83). Figure 8 compares the model with different emissions against observations at the nine AERONET sites in Indochina. FINN1 emissions lead to an improved correlation with observations at all sites and a reduced root mean square model error at six sites compared to GFED3 and GFAS1. Figure 9 compares the multi-annual average seasonal cycle in AOD at four sites in Thailand. The model with GFED3 and GFAS1 emissions underestimates AOD observed during the dry season (~January – May), whereas the model with FINN1 emissions captures the magnitude of dry season AOD reasonably well.

AERONET sites in Indochina (located in north and central Thailand and Vietnam) are influenced by local agricultural burning (Li et al., 2013; Lin et al., 2013; Sayer et al., 2014) of sugarcane and rice crop residues (Gadde et al., 2009; Sornpoon et al., 2014). Agricultural fires are typically smaller than other fire types (e.g., deforestation, grassland/savannah and forest), with burned areas of ~0.3 to ~16 ha reported for individual agricultural fires in the US (McCarty et al., 2009) and Africa (Eva and Lambin, 1998). The prevalence of small fires in Indochina may explain why FINN1 emissions result in better prediction of AOD compared to GFED3 in this region.

We do not find an improved prediction of AOD with GFAS1 compared to GFED3 in this region, although this would be expected since GFAS1 better captures emissions from small fires than GFED3 (Kaiser et al., 2012). However, the GFAS1 FRP is converted to dry matter burned using GFED3 data (Heil et al., 2010; Kaiser et al., 2012), which may lead to an underestimation of small fire emissions in some regions. Conversely, FINN1 assumes a relatively large burned area of 1 km$^2$ (100 ha) for individual agricultural fires and therefore may overestimate emission fluxes in agricultural fire regions. However, since many small fires may be undetected as fire hotspots by MODIS (due to factors such as the small size of the fires, orbital gaps, persistent cloud cover and the timing of satellite overpass i.e. the potential to miss fires events), by oversizing the area of individual burns, the FINN1 emissions may compensate for missing fire detections in this region (B. Yokelson, personal communication, 2014).

## 4.3   Scaling biomass burning emissions

Previous model simulations, summarised in Table 2, underestimate AOD in regions impacted by biomass burning. To improve simulation of AOD, these studies have scaled particulate emissions from biomass burning (or aerosol concentrations) by a factor of 1.02 to 6. We have found that our model with three different fire emission datasets also underestimates both PM2.5

and AOD across tropical regions (although to a lesser extent in Southeast Asia). In this section we explore the impact of scaling biomass burning emissions on simulated AOD and PM2.5 concentrations. We performed two sensitivity simulations with each emission inventory where we perturbed the biomass burning emission fluxes of BC and POM upwards by factors of 1.5 and 3.4 (as recommended for GFED3 and GFAS1 by Kaiser et al. (2012)).

Figures 3b and 3c show the NMBF and correlation between simulated and observed multi-annual monthly mean PM2.5 concentrations for the two simulations with scaled biomass burning emissions. The outcome of scaling the emissions by a factor of 1.5 depends on the site location. At the sites strongly impacted by biomass burning, the model bias in PM2.5 is reduced (FINNx1.5: -0.16 to 0.08; GFEDx1.5: -0.67 to -0.15; GFASx1.5: -0.89 to -0.22) with little change in the correlation. At the preserved forest site near Manaus, the positive model bias is increased (FINNx1.5: 1.33; GFASx1.5: 0.69; GFEDx1.5: 0.66). The outcome of scaling the emissions by a factor of 3.4 depends on both the site location and the emission dataset. The model bias is increased at all sites with FINN1 emissions (0.63-2.72), with mixed results for GFED3 (-0.39 to 1.18) and GFAS1 (-0.16 to 1.25) emissions. Any scaling of the emissions leads to an overestimation of PM2.5 at Manaus with all three emission datasets.

In summary, a scaling factor of 1.5 applied to the FINN1 emissions is adequate for the model to capture surface PM2.5 concentrations observed in regions of high fire activity in the Amazon region. In contrast, the GFAS1 emissions require a larger scaling factor (closer to 3.4) for the model to capture surface PM2.5 observed at these sites.

The results of scaling the GFED3 emissions are more complex. Scaling GFED3 emissions by a factor of 1.5, the model bias becomes relatively small at Alta Floresta (-0.36) and Porto Velho (-0.15) but remains large and negative at Santarem (-0.67). Scaling the emissions by a factor of 3.4 reduces the model bias at Santarem (-0.39), but leads to an overestimation of PM2.5 at the other three sites (0.33-1.18). At Santarem, scaling GFED3 emissions by a factor 3.4 only marginally improves agreement with the observations; the correlation remains below 0.5 and model bias remains negative (despite a positive model bias at the other sites). This is because GFED3 emission fluxes in the peak biomass burning season months in the region of Santarem (November and December) are very low or non-existent, likely due to an omission of small fires (Sect. 4.2), thus there are very few emissions to scale. This result suggests that even by scaling GFED3 emissions by a large factor it is still possible to underestimate PM from fires in regions influenced by emissions from small fires.

Figures 6a and 6b show the NMBF and correlation between simulated and observed multi-annual monthly mean AOD with scaled biomass burning emissions. For the model with GFAS1 emissions, scaling by a factor of 3.4 reduces the model bias at all but one site in Indochina, Africa and South America (relative to the simulations without scaling or with a scaling factor of 1.5), resulting in the best overall match to observed AOD in these regions. In Equatorial Asia the scaling required to capture observed AOD depends on the site location (two sites require no scaling and two sites require a scaling factor of either 1.5 or 3.4).

For GFED3 emissions, scaling by a factor of 3.4 results in the best overall match to observed AOD in Africa and Indochina, but leads to an increased model bias at half the sites in South America. However, even with a scaling factor of 3.4, the model with GFED3 emissions continues to underestimate observed AOD in north Brazil (Belterra; NMBF= -0.94), indicating that a large scaling factor does not fully compensate for the likely omission of small fire emissions in this inventory (Sect. 4.2). The overall result of scaling GFED3 emissions in Equatorial Asia is the same for GFAS1 emissions.

Scaling FINN1 emissions by a factor of 3.4 improves the agreement with observed AOD in Africa (at all sites), but generally leads to overestimation and increased model bias at sites in South America and Southeast Asia. Scaling FINN1 emissions by a factor of 1.5 is adequate to capture observed AOD at the majority of sites in South America (mean NMBF= -0.16), with no scaling required for the majority of sites in Indochina (mean NMBF= 0.02) and Equatorial Asia (mean NMBF= 0.02).

We note that even with a scaling factor of 3.4 applied to the biomass burning emissions, the model underestimates observed AOD at the African AERONET sites with all three fire emission inventories (mean NMBF= -0.31). This may indicate that a larger scaling factor is required to capture observations in this region. However, using a too high scaling factor is likely to compensate for model error e.g. too efficient removal of aerosol or underestimation of dust emissions, and therefore overestimate the contribution of biomass burning to AOD. The potential for compensation errors with emission scaling is relevant for all three regions. For example, in South America the model bias in AOD in the wet season (~December to May) is increased at four or more sites when the FINN1, GFED3, and GFAS1 emissions are scaled by a factor of 3.4, which may be an indication of compensation errors. Compensation errors are also likely to be occurring when emissions are scaled by a factor of 3.4 at sites in urban locations (see Table S1 for location classifications), where a global model is unable to capture sub-grid-scale urban emissions.

# 5. Conclusions

Particulate emissions from open biomass burning (landscape fires) are very uncertain. Numerous previous studies underestimate AOD in regions impacted by fires and scale particulate emissions by up to a factor of 6 to match observed AOD (see Table 2). We have used the GLOMAP global aerosol model evaluated against surface PM2.5 observations and AERONET AOD to better understand particulate emissions from tropical biomass burning.

Simulated AOD is sensitive to a range of variables including i) vertical profile of aerosol, ii) aerosol optical properties, chemical composition, size distribution and hygroscopic growth, iii) relative humidity; and iv) model spatial resolution. In particular, we found that simulated AOD is very sensitive to the calculation of hygroscopic growth, with the magnitude of AOD ranging by a factor of ~1.6 between our upper and lower estimates. We assumed an upper estimate of aerosol hygroscopic growth resulting in an upper estimate of AOD, reducing any emission scaling required to match observed AOD.

We compared three different satellite-derived fire emission datasets (GFED3, GFAS1 and FINN1). Total pan-tropical particulate emission (BC+OC) varied by less than 30% between the different emission datasets. Regional differences were much larger (often exceeding 100%) leading to important differences in aerosol concentrations simulated by the global model.

We found that GLOMAP underestimated both PM2.5 and AOD in regions strongly impacted by biomass burning, with all emission datasets. The largest underestimation of AOD occurred across Africa, which may be partly due to a large contribution of dust. The smallest underestimation of AOD occurred over Equatorial Asia, where the contribution of fire emissions to simulated AOD was also smallest. Overall, the smallest bias between model and both PM2.5 and AOD observations was found using FINN1 emissions. The model with FINN1 emissions also best simulated the seasonal variability of AOD over Indochina, potentially because of the dominance of smaller fires in this region that are better captured by the FINN1 dataset.

In South America where we have coincident surface PM2.5 and AOD observations, underestimation of AOD is greater than underestimation of surface PM2.5, even though we assume upper estimates for aerosol water uptake. We suggest this discrepancy could be caused by errors in the calculation of AOD (see above). We caution against using observations of AOD to scale emissions before these issues are fully understood.

For each emission dataset we ran two additional simulations where we scaled emissions by factors of 1.5 and 3.4, within the range of previous studies (Table 2). We find that the scaling that results in the best agreement with observations is regionally variable and depends on the emission dataset used. With FINN1 emissions, PM2.5 concentrations and AOD in South America are well simulated when emissions are increased by 50%, whereas AOD in Africa is more consistent with a factor 3.4 scaling. In Southeast Asia, observed AOD is well simulated without any scaling applied; scaling FINN1 emissions by 50% generally leads to overestimation in this region. With GFAS1 emissions, PM2.5 concentrations in South America and AOD in South America, Africa and Indochina are best simulated when emissions are scaled by a factor 3.4. With GFED3 emissions, observations of PM2.5 in north Brazil and AOD in Africa, Indochina and some regions of South America are also better simulated with a factor 3.4 scaling; for PM2.5 concentrations and AOD observed in active deforestation regions of South America, a 50% scaling is sufficient. In Equatorial Asia, the results of scaling both GFAS1 and GFED3 emissions are mixed and depend on site location; overall observed AOD is captured best either without scaling or with a scaling factor of 1.5.

A factor 1.5 scaling is within the uncertainty of emission datasets and is substantially smaller than the emission scaling applied by many other studies (see Table 2). We note that a factor 1.5 scaling is within the uncertainty of assumed OM to OC ratios; we assume an OM:OC ratio of 1.4 which is at the low end of other studies (Tsigaridis et al., 2014). We have treated biomass burning emissions as primary and non-volatile. Formation of semi-volatile SOA in biomass burning plumes may be important (Konovalov et al., 2015; Shrivastava et al., 2015) and needs to be explored in future work. Scaling fire emissions by a factor of 3.4 to match AERONET AOD is likely to partly compensate for an underestimation of aerosol from other sources e.g. dust and/or urban emissions and may also compensate for errors in the calculation of AOD. Simulated AOD is sensitive to a range of variables including relative humidity, spatial resolution, aerosol hygroscopicity, aerosol size distribution, aerosol optical properties and mixing state (external/internal mixing). In particular we found strong sensitivity of AOD to aerosol hygroscopicity and water uptake, with simulated AOD at AERONET locations varying by a factor of 1.6 between upper and lower estimates of aerosol hygroscopicity. We note that there is the potential for compensating errors across these different uncertainties. Analysis against detailed aerosol optical, microphysical and chemical measurements (Brito et al., 2014; Andreae et al., 2015) is now required to further understand the issues raised here.

Problems with the detection of small fires are an acknowledged issue for GFED3, which relies on detections of area burned to derive emissions (Randerson et al., 2012). Over regions that are likely dominated by small fires the model with GFED3 emissions substantially underestimates both PM2.5 (north Brazil) and AOD (north Brazil and Thailand). The model with GFAS1 and FINN1 emissions better simulates aerosol in these regions providing independent evidence that these datasets better represent emissions from small fires. We note that the most recent version of GFED emissions (GFED4) includes an estimate of emissions from small fires (Giglio et al., 2013). Future work should evaluate these emissions against aerosol observations to assess the representation of small fire emissions in the specific regions highlighted here.

## Acknowledgements

This research was supported by funding from the Natural Environment Research Council for the South American Biomass Burning Analysis (SAMBBA) project (number NE/J009822/1). The authors gratefully acknowledge the principal investigators (listed in Table S1) and their staff responsible for establishing and maintaining the 27 AERONET stations used in this study and providing quality-assured data.

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

|  | GFED3 | GFAS1 | FINN1 |
|---|---|---|---|
| **Method** | MODIS burned area & biogeochemical model | MODIS thermal anomaly product & fire radiative power | MODIS thermal anomaly product & assumed burned area |
| **Spatial resolution** | 0.5° x 0.5° | 0.5° x 0.5° | 1 km x 1 km |
| **Temporal resolution** | Monthly (1997 – 2011) Daily (2003 – 2011) | Daily (2001 – 2015) | Daily (2002 – 2013) |
| **Amount of OC emitted over tropics (Tg a$^{-1}$)** | 13.412 | 11.731 (0.87) | 17.282 (1.29) |
| **Amount of BC emitted over tropics (Tg a$^{-1}$)** | 1.705 | 1.532 (0.90) | 1.724 (1.01) |
| **OC:BC ratio over tropics** | 7.87 | 7.66 | 10.02 |
| **Reference** | Van der Werf et al., 2010 | Kaiser et al., 2012 | Wiedinmyer et al., 2011 |

**Table 2.** Summary of scaling factors applied in previous modelling studies to biomass burning emissions or modelled concentrations of biomass burning aerosol to match observations. Region abbreviations used in the table are defined in van der Werf et al. (2006): Northern Hemisphere South America (NHSA), Southern Hemisphere South America (SHSA), Northern Hemisphere Africa (NHAF), Southern Hemisphere Africa (SHAF), Southeast Asia including the Philippines (SEAS) and Equatorial Asia (EQAS). See van der Werf et al. (2006; 2010) for discussion of differences between GFED versions 1, 2 and 3; on average GFED3 are 13% lower than GFED2 van der Werf et al. (2010), with total GFED2 emissions lower than GFED1 in Central and Southern America and Southern Africa (van der Werf et al., 2006).

| Reference | Biomass burning emission inventory | Region of focus | Details of scaling applied |
|---|---|---|---|
| Myhre et al., 2003 | Biomass burning BC emissions from the Global Emissions Inventory Activity (GEIA), based on Cooke and Wilson (1996); OC emissions from Liousse et al. (1996). | Southern Africa | Used a relatively high OM/OC ratio of 2.6 and increased the modelled aerosol mass by 20% to account for mass fraction of inorganic components observed to be of 17% of the total mass. |
| Matichuk et al., 2007 | GFED1 (van der Werf et al., 2003) | Southern Africa | Multiple sensitivity studies were performed with the model including simulations with halved and doubled fire emissions.. |
| Matichuk et al., 2008 | GFED2 (van der Werf et al., 2006) | South America | Smoke source function was scaled up by a factor of 6.. |
| Johnson et al., 2008 | Biomass burning emissions following Dentener et al. (2006): GFED1 (van der Werf et al., 2004) for year 2000 or a 5-year (1997–2001) average (not specified) | West Africa | Increased mass concentration of biomass burning AOD by a factor of 2.4.. |
| Chin et al., 2009 | Calculated using dry mass burned dataset from GFED2 (van der Werf et al., 2006) | Global | No scaling applied, but used emission factors of BC (1 g kg$^{-1}$) and OC (8 g kg$^{-1}$) that are 40–100% higher than commonly used values (Andreae and Merlet, 2001). |
| Sakaeda et al., 2011 | Aerosol fields taken from MATCH chemical transport model | Southern Africa | OC and BC masses were increased by a factor of 2 over 10°N–30°S and 20°W–50°E. |
| Johnston et al., 2012 | GFED2 (van der Werf et al., 2006) | Global | Scalar adjustments made for 14 continental scale regions: NHSA (2.48-2.7), SHSA (1.9-3.3), NHAF (1.02-1.08), SHSA (1.68-2.01), SEAS (2.43-3.08), EQAS (2.3-2.72). Scaling factors were applied to modelled surface fire PM2.5 to match satellite observations of AOD (non-fire aerosol was also scaled). |
| Kaiser et al., 2012 | GFED3 and GFASv1.0 | Global | Model was biased low in South America and Africa by factors of 4.1 and 3.0. Recommended a global |

| | | | enhancement of 3.4 for PM emissions from fires. |
|---|---|---|---|
| Ward et al., 2012 | Calculated from Kloster et al. (2010, 2012) CLM3 simulations of global fire area burned; using emission factors from Andreae and Merlet (2001) and updates from Hoelzemann et al. (2004). Compared against GFED2. | Global | Scalar adjustments were made for continental scale regions following Johnston et al. (2012) with slight modifications: SHSA (2.0), NHAF (1.0), SHAF (3.0), SEAS (1.5), EQAS (3.0). Scaling factor directly applied to model fire emissions. |
| Tosca et al., 2013 | GFED3 | Global | Biomass burning BC and OC emissions scaled by factor of 2 globally with additional regional scaling factors applied: South America (2.4), Africa (2.1), Southeast Asia (1.67). |
| Marlier et al., 2013 | GFED3 | Southeast Asia | Total aerosol burden scaled by 1.02-1.96 (depending on model), with additional scaling factors of 1.36-2.26 applied to fire aerosol.. |

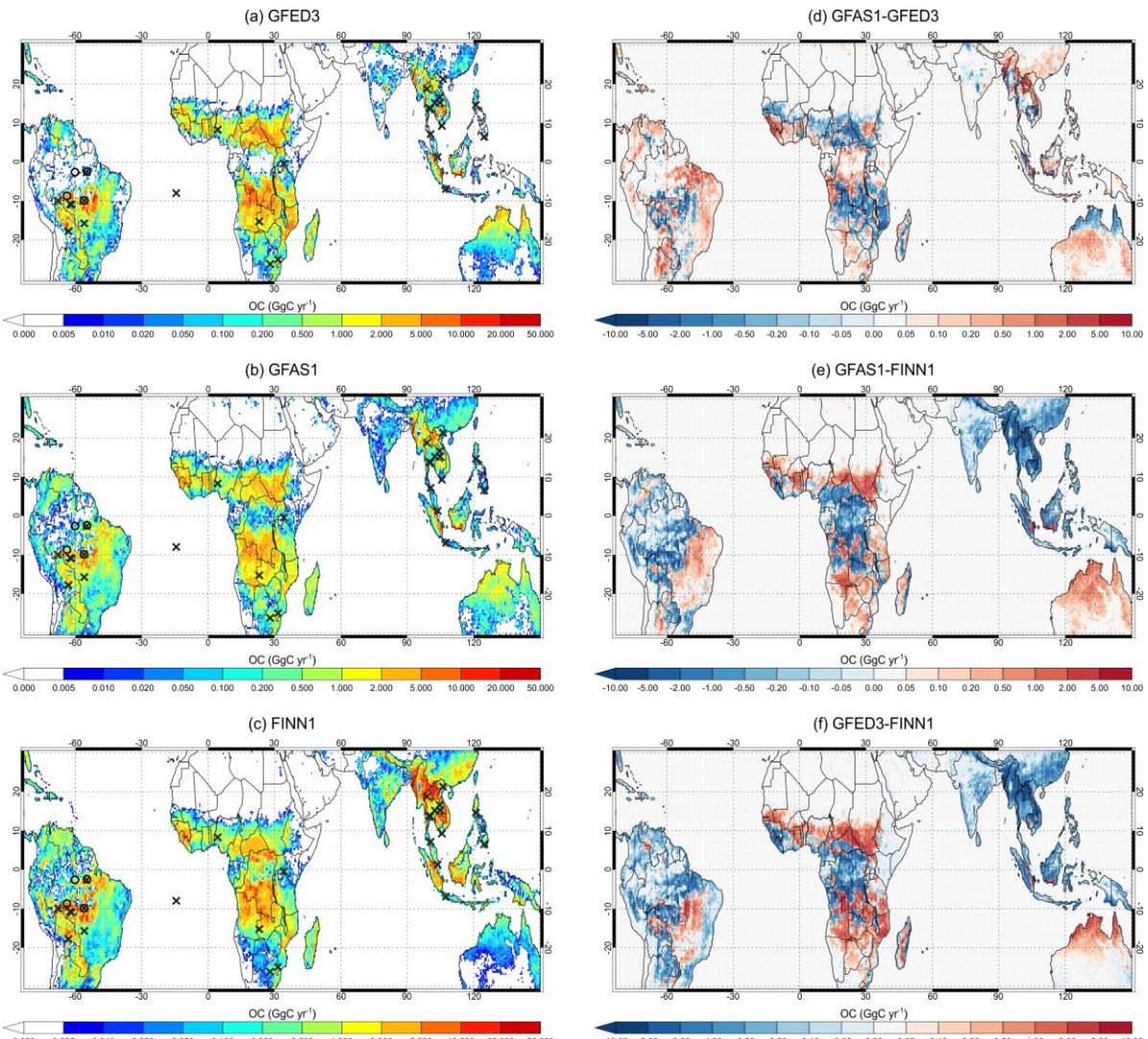

**Figure 1.** (a)-(c) Total annual emissions of organic carbon (OC) in Gg(C) a⁻¹ averaged over the period of January 2003 to December 2011 from **(a)** GFED3, **(b)** GFAS1 and **(c)** FINN1. Black circles mark the locations of the four aerosol measurement stations and black crosses mark the locations of the 27 AERONET stations (see Table S1). (d)-(f) Absolute difference in 2003-2011 mean annual OC emissions between GFAS1, GFED3 and FINN1 **(d)** GFAS1 minus GFED3 **(e)** GFAS1 minus FINN1 **(f)** GFED3 minus FINN1. The FINN1 OC emissions (with a 1 km x 1 km horizontal resolution) were aggregated onto a grid of 0.5° x 0.5° degree resolution to compare with GFED3 and GFAS1.

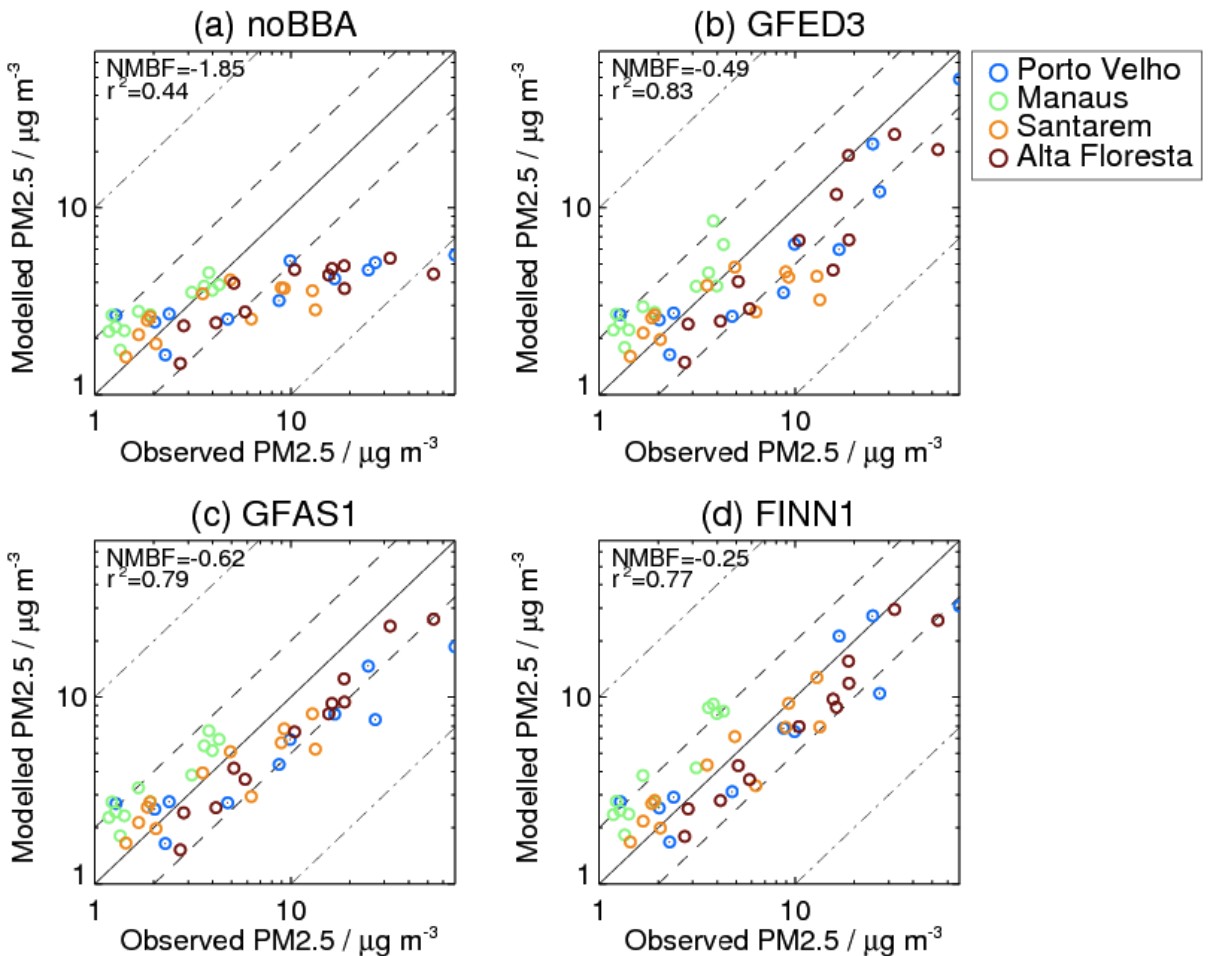

**Figure 2.** Simulated versus observed multi-annual monthly mean PM2.5 concentrations at each ground station in the Amazon region for the model **(a)** without biomass burning emissions, and with **(b)** GFED3, **(c)** GFAS1 and **(d)** FINN1 emissions. Multi-annual monthly mean concentrations were calculated by averaging over all years of data available between January 2003 and December 2011 to obtain an average seasonal cycle at each station. The normalised mean bias factor (NMBF; Yu et al., 2006) and Pearson's correlation ($r^2$) between modelled and observed PM2.5 concentrations are shown in the top left corner.

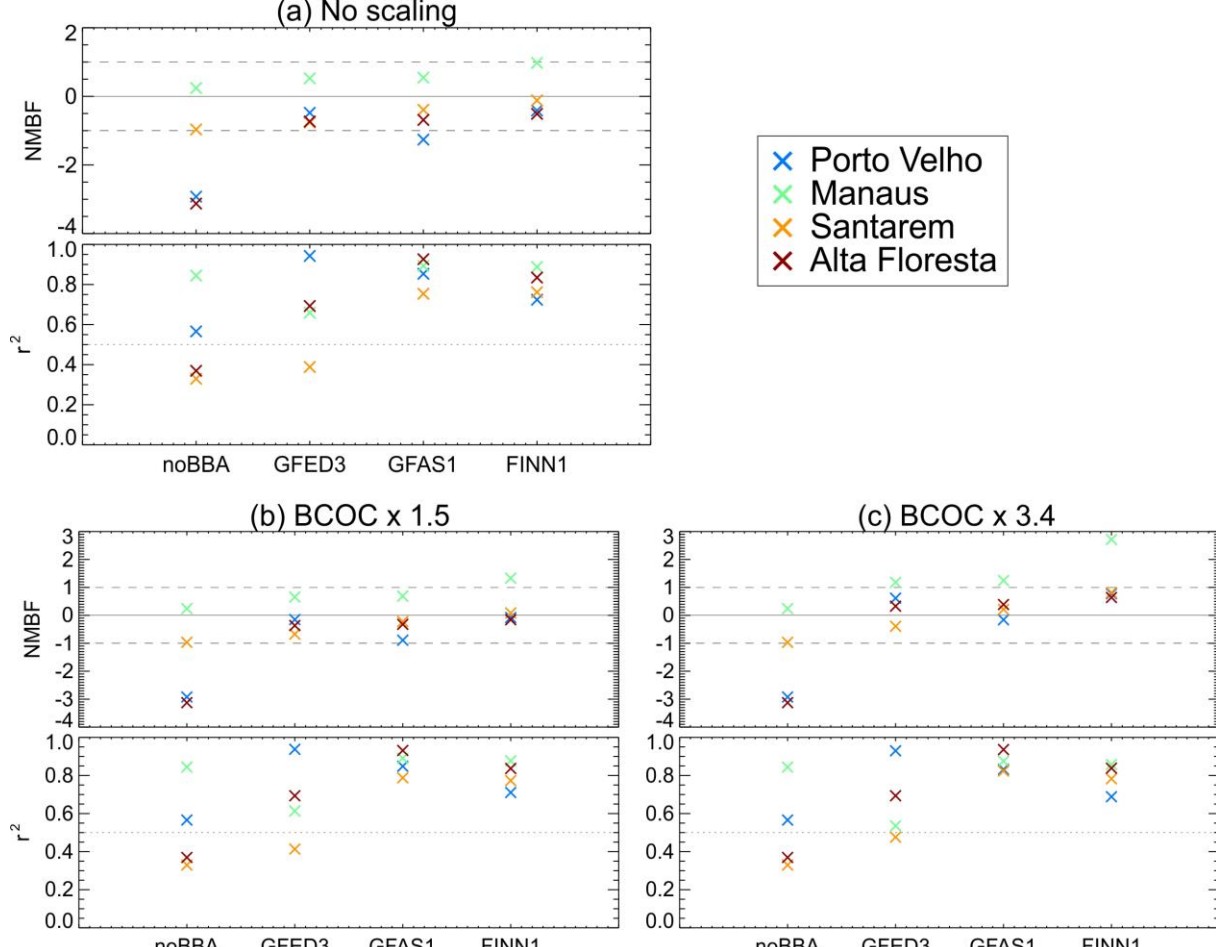

**Figure 3.** Normalised mean bias factor (NMBF; Yu et al., 2006) and Pearson's correlation coefficient ($r^2$) between modelled and observed multi-annual monthly-mean PM2.5 concentrations at each of the four ground stations in Amazonia. Results are shown for four model simulations: without fires (noBBA), and with each of the three biomass burning emissions inventories: GFED3, GFAS1, FINN1. (a) No scaling applied to the fire emissions; (b) particulate (BC/OC) fire emissions scaled up globally by a factor 1.5; (c) particulate (BC/OC) fire emissions scaled up globally by a factor of 3.4. The dashed lines indicate NMBFs of -1 and 1, which equate to an underestimation and overestimation, respectively, of a factor of 2. The dotted line indicates an $r^2$ value of 0.5.

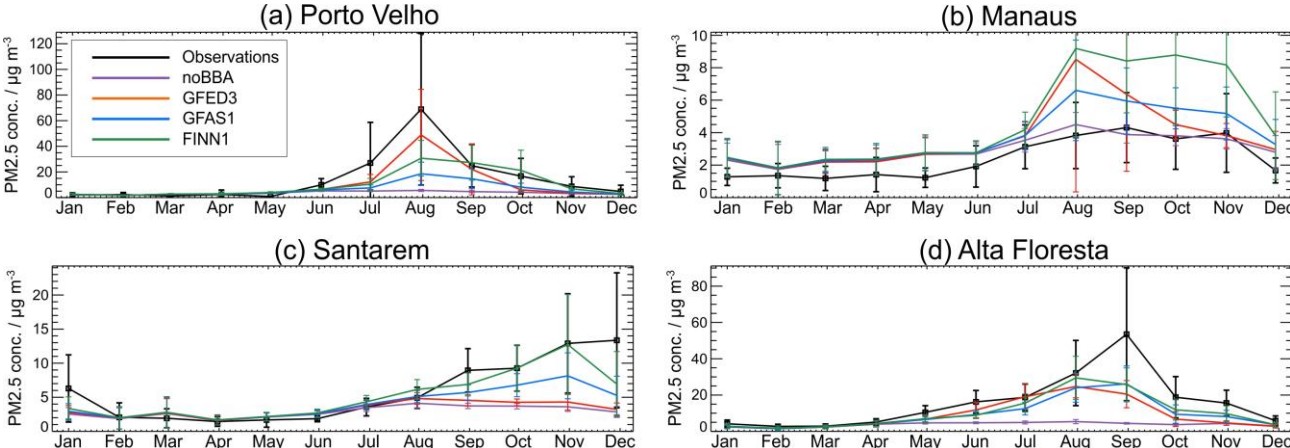

**Figure 4.** Average seasonal cycles in observed (black) and simulated (colour) multi-annual monthly mean PM2.5 concentrations at four ground stations in the Amazon region: **(a)** Porto Velho (2009-2011); **(b)** Manaus (2008-2011); **(c)** Santarem (2003-2006); and **(d)** Alta Floresta (2003-2004). Multi-annual monthly mean concentrations were calculated by averaging over all years of available observation data between January 2003 and December 2011. The modelled results are shown for four simulations: without biomass burning (purple), with GFED3 emissions (red), with GFAS1 emissions (blue) and with FINN1 emissions (green). The error bars show the standard deviation of the mean of the observed and simulated values, which represents the inter-annual and intra-monthly variability in the daily mean PM2.5 concentrations.

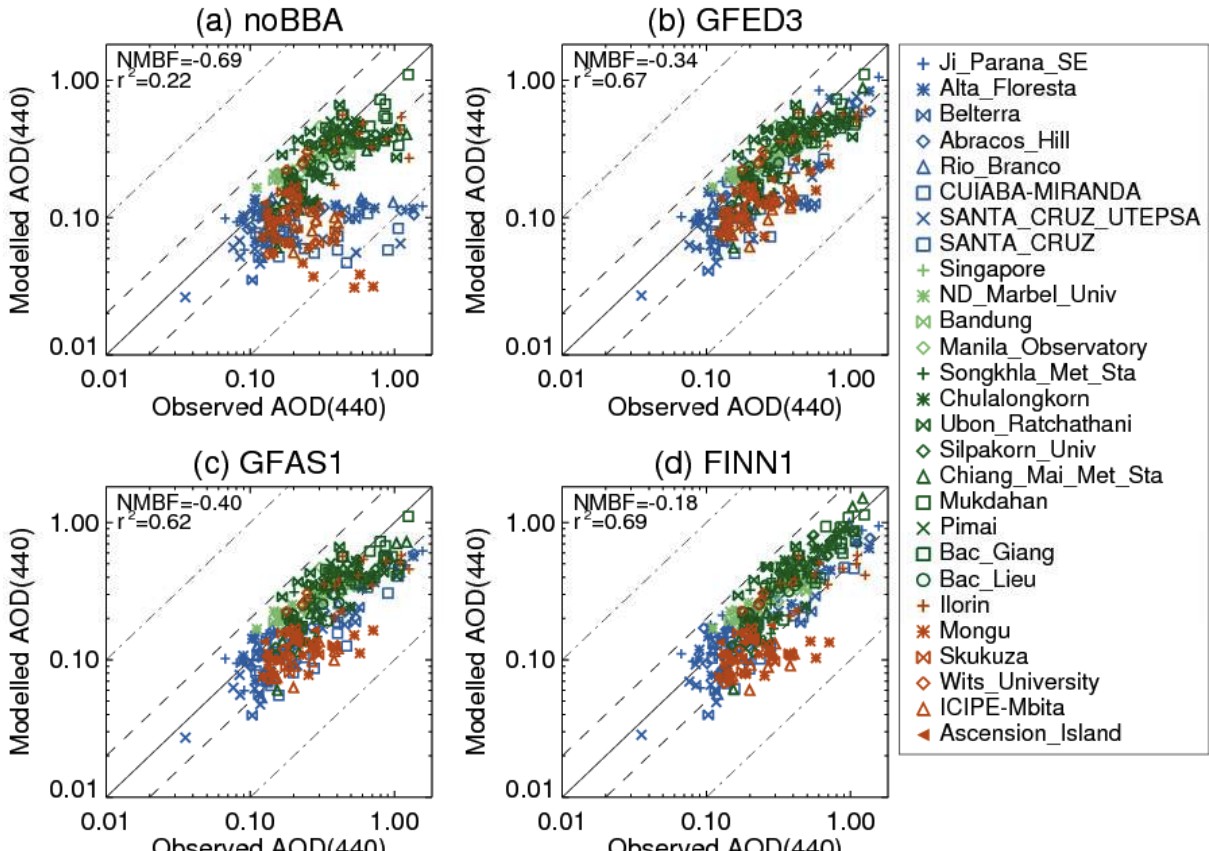

**Figure 5.** Simulated versus observed multi-annual monthly mean AOD at 440 nm at each AERONET station. The model is shown **(a)** without biomass burning emissions, and with **(b)** GFED3, **(c)** GFAS1 and **(d)** FINN1 emissions. As for Fig. 2, the multi-annual monthly mean AODs were calculated using all years of daily mean data available between January 2003 and December 2011 to obtain an average seasonal cycle at each station. AERONET stations located in South America are shown in blue; stations in Southeast Asia are shown in green (stations in Equatorial Asia and Indochina in light and dark green, respectively); and stations in Africa are shown in orange. The normalised mean bias factor (NMBF) and Pearson's correlation ($r^2$) between modelled and observed PM2.5 concentrations are shown in the top left corner.

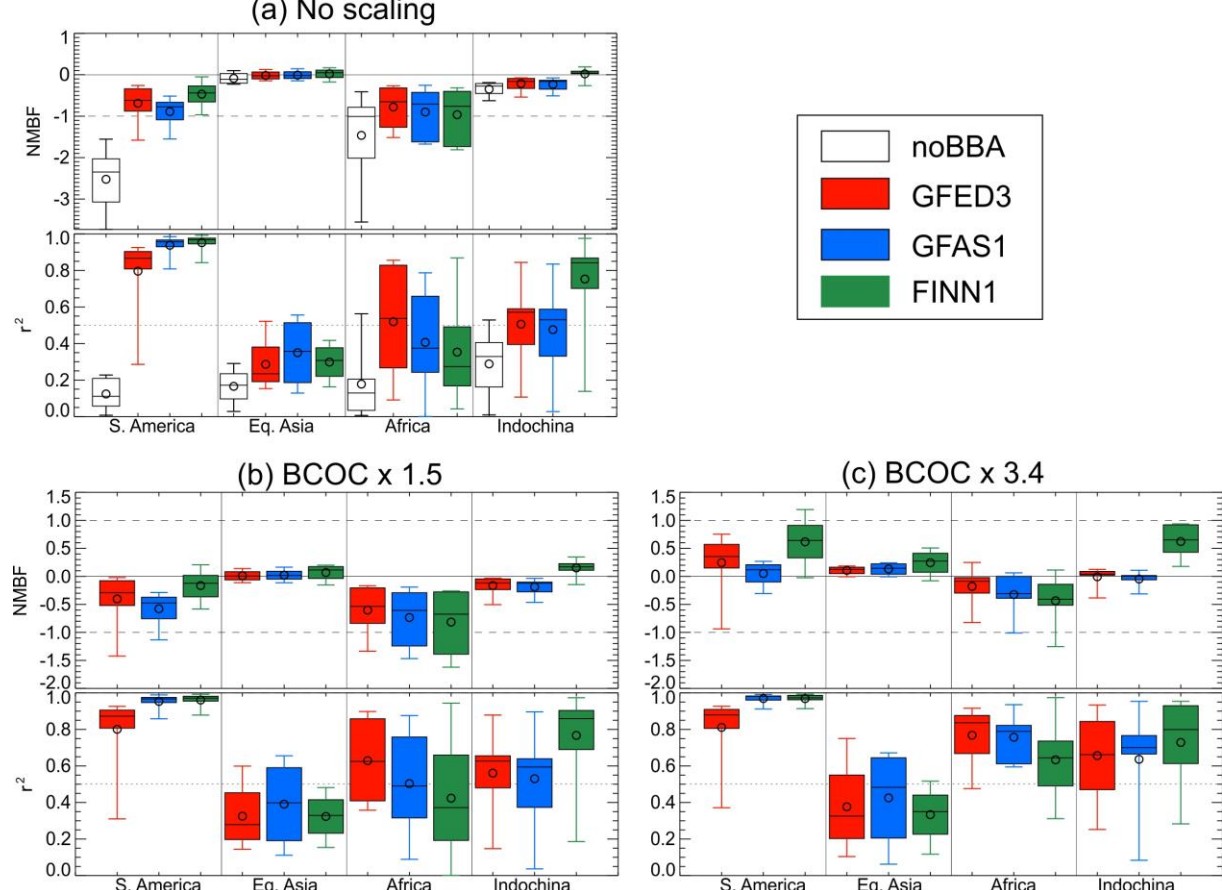

**Figure 6.** Box and whisker plots of the normalised mean bias factor (NMBF) and Pearson's correlation coefficient ($r^2$) between modelled and observed multi-annual monthly-mean AOD at 440 nm for AERONET stations located in South America (8 sites), Equatorial Asia (4 sites), Africa (6 sites) and Indochina (9 sites). Results are shown for four model simulations: without fires (white), and with each of the three biomass burning emissions inventories: GFED3 (red), GFAS1 (blue), FINN1 (green). (a) No scaling applied to the fire emissions; (b) particulate (BC/OC) fire emissions scaled up globally by a factor 1.5; (c) particulate (BC/OC) fire emissions scaled up globally by a factor of 3.4. The dashed lines indicate NMBFs of -1 and 1, which equate to an underestimation and overestimation, respectively, of a factor of 2. The dotted line indicates an $r^2$ value of 0.5.

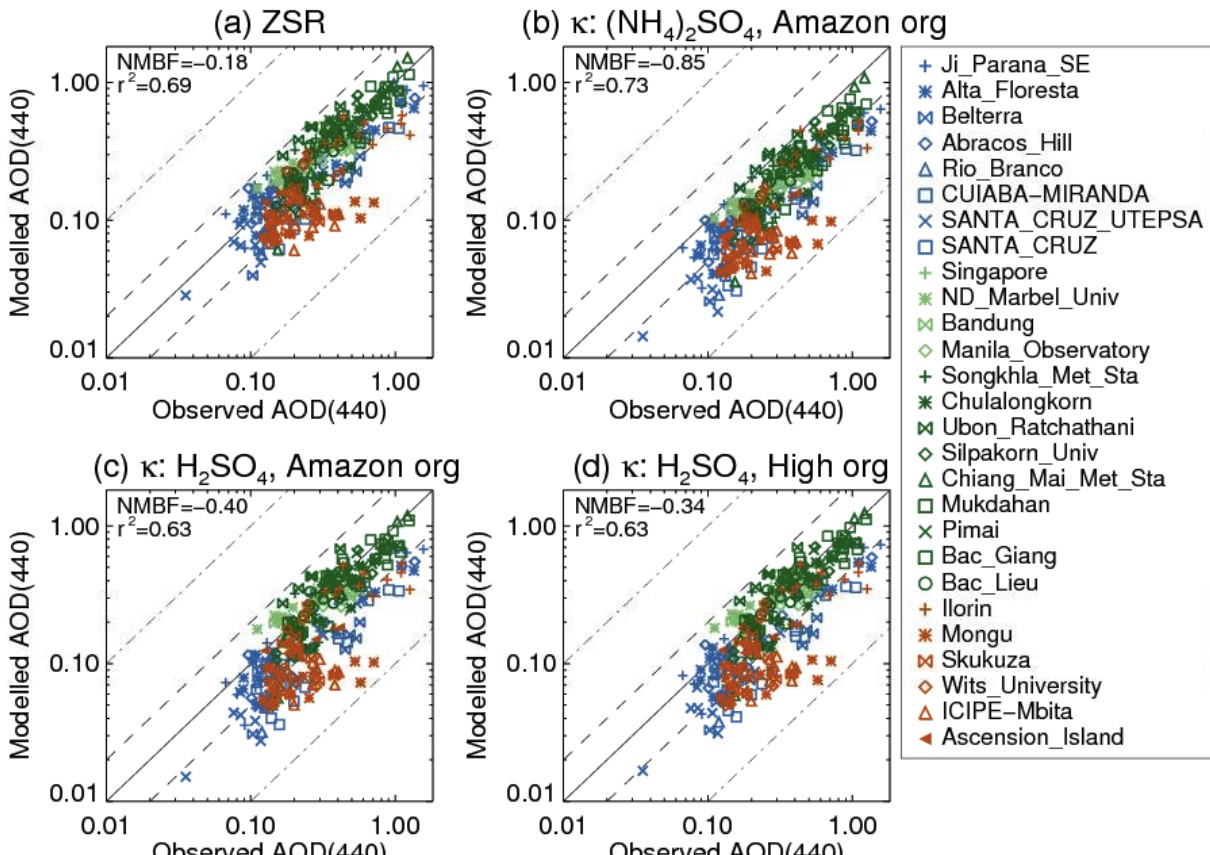

**Figure 7.** Simulated versus observed multi-annual monthly mean AOD at 440 nm at each AERONET

station to demonstrate the sensitivity of simulated AOD to the calculation of aerosol water uptake. The

model is with FINN1 fire emissions and simulated AOD is calculated assuming internal mixing with **(a)**

ZSR water uptake scheme (identical to Fig. 5d); **(b)** κ-Köhler water uptake scheme: $\kappa_{SO4}$=0.53, $\kappa_{POM}$=0.1; **(c)** κ-Köhler water uptake scheme: $\kappa_{SO4}$=1.19, $\kappa_{POM}$=0.1; and **(d)** κ-Köhler water uptake

scheme: $\kappa_{SO4}$=1.19, $\kappa_{POM}$=0.2. AERONET stations located in South America are shown in blue;

stations in Southeast Asia are shown in green (stations in Equatorial Asia and Indochina in light and

dark green, respectively); and stations in Africa are shown in orange. The normalised mean bias factor

(NMBF) and Pearson's correlation ($r^2$) between modelled and observed PM2.5 concentrations are shown

in the top left corner.

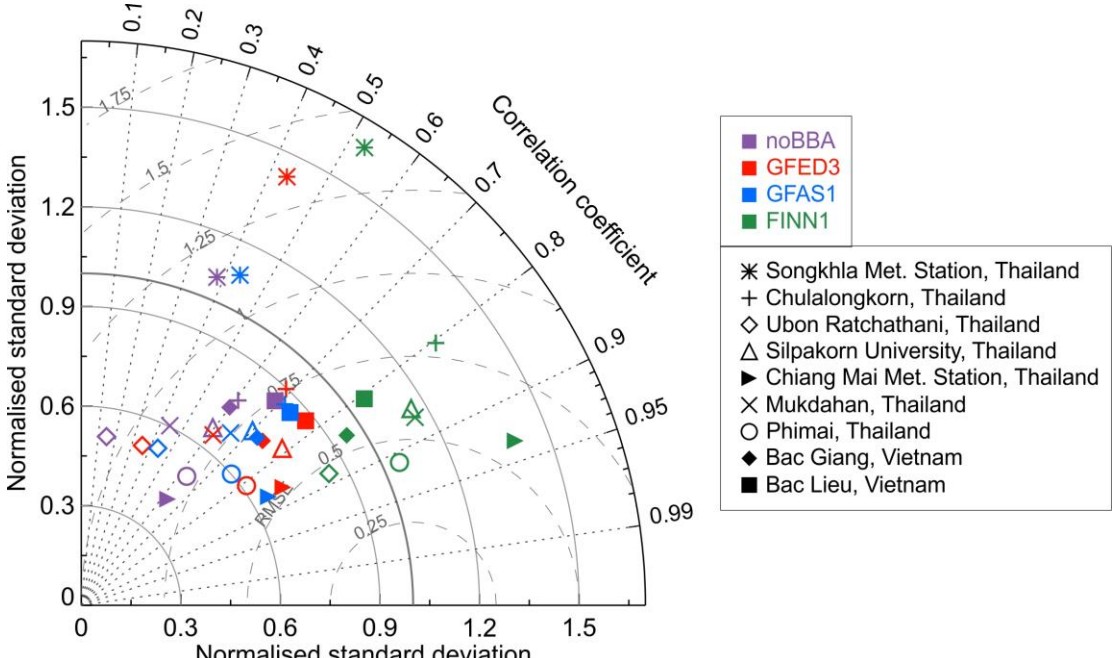

**Figure 8.** Taylor diagrams (Taylor, 2001) comparing monthly mean modelled and observed AOD (440 nm) at 9 AERONET stations located in Indochina. The modelled and observed monthly mean AODs were calculated for every month with available daily mean data between January 2003 and December 2011. The observations are represented by a point on the x-axis at unit distance from the y-axis. The results are shown for four simulations: without biomass burning (purple), and with GFED3 (red), GFAS1 (blue) and FINN1 (green) fire emissions. The model standard deviation and root mean square error (RMSE) are normalised by dividing by the corresponding observed standard deviation. The normalised standard deviation and RMSE values are marked by the grey-solid and grey-dashed lines respectively. The correlation coefficient (r) values are marked by the grey dotted lines.

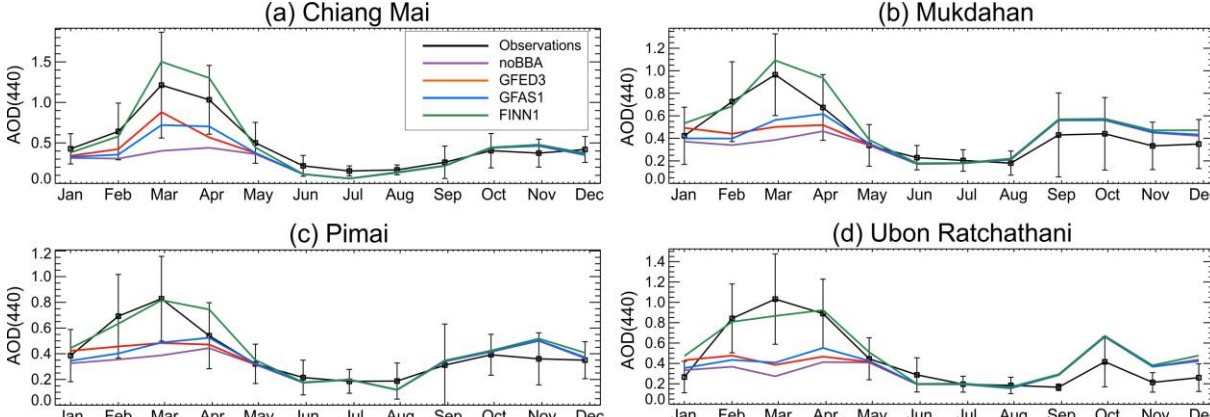

Figure 9. Average seasonal cycles in observed (black) and simulated (colour) monthly mean AOD at 440 nm at four AERONET stations in the Thailand: **(a)** Chiang Mai Met. Station; **(b)** Mukdahan; **(c)** Phimai; and **(d)** Ubon Ratchathani. Multi-annual monthly mean concentrations were calculated by averaging over all years of available daily mean observation data between January 2003 and December 2011. The modelled results are shown for four simulations: without biomass burning (purple), and with GFED3 (red), GFAS1 (blue) and FINN1 (green) fire emissions. The error bars show the standard deviation of the mean of the observations.