# Peer review of "Analysis of particulate emissions from tropical biomass"

_Atmospheric Chemistry and Physics, 2015_

## Referee Comment (RC1) · Anonymous Referee #1 · 14 Feb 2016

The paper of Reddington et al. investigates the impacts of biomass burning on tropical aerosols. This is done with GLOMAP global aerosol model, evaluated by long-term surface observation of PM2.5 and AOD. Specifically, this work compares three different fire emission datasets (GFED3, GFAS1 and FINN) to explore the uncertainty in emissions.

This study aims to "better understand the discrepancy in modeled biomass burning AOD and to ultimately improve estimates of biomass burning aerosol". While the authors address the contribution of underestimation in biomass burning aerosols to the bias in AOD, it would have been beneficial if they could perform further analysis to see the relative contribution of other factors. For example, if they assume internal mix-

ing instead of external mixing, what does this do to the modeled AOD bias? Is the uncertainty in RH large enough to explain the bias in modeled AOD?

In summary, this paper is well written. It describes what they did and is easy to follow along. It adds value to the literature on this topic and is worthy of publication in ACP subject to addressing these.

Other minor things:

1. Are the model results also obtained for the year of 2003-2011? I did not see it in the text.

2. I found it difficult to read the tiles for x/y axis and legend in figure 4&8.

---

## Referee Comment (RC2) · Anonymous Referee #2 · 15 Feb 2016

General comments

The paper evaluates multi-annual (2003-2011) monthly mean values of near-surface aerosol mass concentration (PM2.5) and aerosol optical depth (AOD) simulated by the GLOMAP global aerosol model against corresponding measurements conducted at four ground stations in the Amazon region and at 27 AERONET stations located in tropical regions worldwide. The simulations were done with three different datasets of biomass burning emissions (GFED3, FINN1, and GFAS1). Additional numerical experiments involved scaling the biomass burning emissions by a factor of 1.5 or 3.4. The model performance is evaluated in terms of the Pearson correlation coefficient and the normalized mean bias factor (NMBF). It is found that the model considerably

underestimates both PM2.5 concentrations and AOD, with a greater underestimation of AOD than PM2.5. The paper is well written and the presentation quality is good. However, the scientific significance and the overall scientific quality of the study are questionable.

Indeed, the fact that models tend to underestimate AOD over regions affected by fires (including the Amazon region) is well known. This is acknowledged by the authors: some earlier studies reporting the underestimation of AOD are mentioned in the paper, although the list of such studies is certainly not complete (see also, e.g., Petrenko et al., 2012; Konovalov et al., 2014). The use of ground measurements of surface aerosol mass concentration together with AOD measurements is a relatively novel point. However, the parallel analysis of PM2.5 and AOD measurements, as well as the use of three different emission datasets also did not help to fully explain the mismatch between the model and measurements of AOD. Note that Petrenko et al. (2012) has presented a much more extensive analysis of the impact of biomass burning emission uncertainties on simulated AOD values.

A serious drawback of the analysis is that it is based on an obsolete / simplistic understanding of organic aerosol processes. In particular, the authors disregard the well established facts that organic aerosol (which is a major fraction of biomass burning aerosol) is formed by organic compounds featuring a broad distribution of volatilities (see, e.g., May et al., 2013) and that a part of them, while in the gas phase, can provide a major source of secondary organic aerosol (SOA) (see, e.g., Grieshop et al., 2009; Hennigan et al., 2011) as a result of oxidation processes. Meanwhile, these facts have direct implications for biases in biomass burning aerosol emission inventories and for the mismatch between simulations and measurements of AOD (Jathar et al., 2014; Konovalov et al., 2015; Shrivastava et al., 2015). Although it is briefly mentioned that the SOA formation in biomass burning plumes can contribute to the difference between the simulations and observations, any quantitative estimates of such a contribution are not provided. A study could benefit from simulations of biomass burning aerosol by

using the volatility basis set framework and available parameterizations (e.g., Hodzic at al., 2010; Shrivastava et al., 2013, 2015). In any case, simplifications made in the model in regard to organic aerosol processes (such as an implicit assumption that organic part of biomass burning aerosol consist only of non-volatile material) and their implications for the results of this analysis had to be carefully described and discussed in light of earlier findings from relevant laboratory, field and modeling studies.

Specific comments

1. Page 6. The PM2.5 measurements made by the gravimetric filter analysis method that is known to be associated with large uncertainties (Malm et al., 2011). The authors estimate uncertainties of such measurements to be 15%. But how was the loss of organic aerosol mass due to desorption estimated? Available volatility distributions of fresh biomass burning emissions (e.g., May et al., 2013) imply that the loss of the organic aerosol mass from samples taken inside biomass burning plumes (POM$\sim$1000 ug/m3) after equilibration to ambient conditions (POM$\sim$10 ug/m3) could be as large as 40 percent.

2. Page 9, line 12. The fire emissions were injected into the model by using a set of fixed ecosystem-dependent altitudes. Meanwhile, it is known that the injection height depends on the fire intensity. If, for example, the injection height for major fires was underestimated in the model, the surface PM2.5 concentration during fire seasons could be overestimated. The study could benefit from using one of more realistic parameterizations of the injection height (e.g., Sofiev et al., 2012; Paugam et al., 2016). And, anyway, it would be important to ensure by means of a sensitivity analysis that the discrepancies between the results obtained with PM2.5 and AOD measurements are not due to biases in the injection height. The adequacy of the injection heights could further be evaluated by using surface measurements of CO concentrations and satellite observations of CO columns (see, e.g., Konovalov et al., 2014).

3. Page 10, line 13. "The water uptake for each soluble aerosol component is calculated on-line in the model according to ZSR theory". Was hygroscopicity of organic components of biomass burning aerosol taken into account in the simulations? If so, what were typical values of the hygroscopic growth factor for the organic fraction?

4. Section 3.3. The paper could significantly benefit from an analysis of inter-annual variability of fire emissions and of corresponding PM2.5/AOD values in the Amazon region during the fire season. Has such a variability been predicted by the different inventories consistently? Can the model reproduce the observed inter-annual variability in PM2.5? Which of the inventories considered does enable the best agreement between the inter-annual variations in the simulations and measurements of PM2.5?

5. Page 15, line 17. "This suggests that the negative model bias in the dry season is largely due to uncertainty in the biomass burning emissions rather than anthropogenic emissions, SOA or microphysical processes in the model." Please see above a general comment about the potential importance of SOA formation in biomass burning plumes.

6. Page 20, line 20. "Uncertainties exist in the calculation of AOD that may contribute to the negative bias in simulated AOD." Did the authors try to validate their AOD calculations with other independent data? For example, it would be interesting to see if the model calculations are consistent with available measurements of the mass scattering and absorbing efficiencies (e.g. Reid et al., 2005). A bias in these parameters would indicate a similar bias in the AOD calculations.

Minor comments

1. Page 11, line 17. Daily GFED3 fire emissions were implemented in GLOMAP for the period 2003–2011, with monthly emissions implemented for the period 1997–2002. Were simulations analyzed in this paper really extended to the period 1997–2002?

2. Several papers cited in the text (Chin et al., 2009; Randerson et al., 2012; Zhou et al. 2002 . . . ) are missing in the references.

References:

Grieshop, A. P., Logue, J. M., Donahue, N. M., and Robinson, A. L.: Laboratory investigation of photochemical oxidation of organic aerosol from wood fires 1: measurement and simulation of organic aerosol evolution, Atmos. Chem. Phys., 9, 1263–1277, doi:10.5194/acp-9-1263-2009, 2009.

Hennigan, C. J., Miracolo, M. A., Engelhart, G. J., May, A. A., Presto, A. A., Lee, T., Sullivan, A. P., McMeeking, G. R., Coe, H., Wold, C. E., Hao, W.-M., Gilman, J. B., Kuster, W. C., de Gouw, J., Schichtel, B. A., Collett Jr., J. L., Kreidenweis, S. M., and Robinson, A. L.: Chemical and physical transformations of organic aerosol from the photo-oxidation of open biomass burning emissions in an environmental chamber, Atmos. Chem. Phys., 11, 7669–7686, doi:10.5194/acp-11-7669-2011, 2011.

Hodzic, A., Jimenez, J. L., Madronich, S., Canagaratna, M. R., DeCarlo, P. F., Kleinman, L., and Fast, J.: Modeling organic aerosols in a megacity: Potential contribution of semi-volatile and intermediate volatility primary organic compounds to secondary organic aerosol formation, Atmos. Chem. Phys., 10, 5491–5514, doi:10.5194/acp-10-5491-2010, 2010.

Konovalov, I. B., Berezin, E. V., Ciais, P., Broquet, G., Beekmann, M., Hadji-Lazaro, J., Clerbaux, C., Andreae, M. O., Kaiser, J. W., and Schulze, E.-D.: Constraining CO2 emissions from open biomass burning by satellite observations of co-emitted species: a method and its application to wildfires in Siberia, Atmos. Chem. Phys., 14, 10383–10410, doi:10.5194/acp-14-10383-2014, 2014.

Konovalov, I. B., Beekmann, M., Berezin, E. V., Petetin, H., Mielonen, T., Kuznetsova, I. N., and Andreae, M. O.: The role of semi-volatile organic compounds in the mesoscale evolution of biomass burning aerosol: a modeling case study of the 2010 mega-fire event in Russia, Atmos. Chem. Phys., 15, 13269-13297, doi:10.5194/acp-15-13269-2015, 2015.

May, A. A., Levin, E. J. T., Hennigan, C. J., Riipinen, I., Lee, T., Collett Jr., J. L., Jimenez, J. L., Kreidenweis, S. M., and Robinson, A. L.: Gas-particle partitioning of

primary organic aerosol emissions: 3. Biomass burning, J. Geophys. Res. Atmos., 118, 11327–11338, doi:10.1002/jgrd.50828, 2013.

Jathar, S. H., Gordon, T.D., Hennigan, C.J., Pye, H. O. T., Pouliot, G., Adams P. J., Donahue N. M., and Robinson A.L.: Unspeciated organic emissions from combustion sources and their influence on the secondary organic aerosol budget in the United States, PNAS, 111, 10473–10478, doi: 10.1073/pnas.1323740111, 2014.

Malm, W.C., Schichtel, B.A. and Pitchford, M. L., Uncertainties in PM2.5 gravimetric and speciation measurements and what we can learn from them, Journal of the Air & Waste Management Association, 61(11), 1131-1149, doi:10.1080/10473289, 2011.

Paugam, R., Wooster, M., Freitas, S., and Val Martin, M.: A review of approaches to estimate wildfire plume injection height within large-scale atmospheric chemical transport models, Atmos. Chem. Phys., 16, 907-925, doi:10.5194/acp-16-907-2016, 2016.

Petrenko, M., Kahn, R., Chin, M., Soja, A., Kucsera, T., and Harshvardhan: The use of satellite-measured aerosol optical depth to constrain biomass burning emissions source strength in the global model GOCART, J. Geophys. Res., 117, D18212, doi:10.1029/2012JD017870, 2012.

Reid, J. S., Eck, T. F., Christopher, S. A., Koppmann, R., Dubovik, O., Eleuterio, D. P., Holben, B. N., Reid, E. A., and Zhang, J.: A review of biomass burning emissions part III: intensive optical properties of biomass burning particles, Atmos. Chem. Phys., 5, 827–849, doi:10.5194/acp-5-827-2005, 2005. Sofiev, M., Ermakova, T., and Vankevich, R.: Evaluation of the smoke-injection height from wild-land fires using remote-sensing data, Atmos. Chem. Phys., 12, 1995–2006, doi:10.5194/acp-12-1995-2012, 2012.

Shrivastava, M., Zelenyuk, A., Imre, D., Easter, R.C., Beranek„J., Zaveri, R.A, and Fast, J.D.: Implications of Low Volatility and Gas-phase Fragmentation Reactions on SOA Loadings and their Spatial and Temporal Evolution in the Atmosphere. J. Geophys.
[Figure]

Res. Atmos., 118(8), 3328-3342, doi: 10.1002/jgrd.50160, 2013.

Shrivastava, M., Easter, R., Liu, X., Zelenyuk, A., Singh, B., Zhang, K., Ma, P-L, Chand, D., Ghan, S., Jimenez, J.L., Zhang, Q., Fast, J., Rasch, P. and Tiitta, P.: Global transformation and fate of SOA: Implications of low volatility SOA and gas-phase fragmentation reactions, J. Geophys. Res. Atmos., 120, 4169–4195, doi:10.1002/2014JD022563, 2015.

---

## Referee Comment (RC3) · Anonymous Referee #3 · 4 Mar 2016

This manuscript evaluates global aerosol model simulations that have been performed with the GLOMAP model and three widely used fire emission inventories, namely GFED3, GFAS1 and FINN1. The simulations are validated thoroughly and in considerable detail with AOD and PM2.5 measurements performed in the tropics, i.e. South America, Africa and SE Asia. The study addresses the most pertinent issues recently discussed in the field of smoke aerosol modelling, i.e. the omission of small fires in burnt-area-based inventories and the need to scale up the pyrogenic aerosol flux for use in global atmospheric models. The statistical analysis is based on monthly mean values. The study is therefore very well suited as a guide on how to best select one of the fire emission inventories for use with GLOMAP, and on how accurate the simulated

smoke AOD and PM2.5 may be. Considering the wide use of GLOMAP and of the investigated emission inventories, the study presents relevant results that are worth publication in ACP.

The study is well written and clearly presented. It adds quantitative detail to the already existing characterization of the fire emission inventories. However, this quantitative detail appears to be linked to using the GLOMAP model, and it cannot necessarily be transferred to use in other atmospheric aerosol models. The authors missed several opportunities to obtain more generally applicable new results. In particular:

- Correlations are calculated from monthly averages like so many studies have done in the past. Since emission, model and observation data are available with daily resolution, investigating this time scale would have been easily possible and and much more novel.

- The study shows that PM2.5 and AOD require different upscaling of emissions. It would have been most interesting and new to study possible reasons for this. I suspect, it points to model shortcomings, but in which part of the model?

- Likewise, it would have been of general interest to see whether any of the model configuration parameters have an impact on the amount of upscaling required for any given inventory.

I am aware that addressing one of these issues in the final manuscript will imply a major effort, which may not be justified at this stage. However, if the authors would be willing to do it, this would certainly make the results applicable for a much larger community, i.e. also those who use other models than GLOMAP or its results.

Since the manuscript is very well written, I have only very few minor comments:

SPECIFIC COMMENTS

p.11, l.1 and p.12, l.13: delete "yearly varying"

p.11, l.28: You may cite Seiler  Crutzen 1980 for this formula.

p.14, l.16: Please add the definition of NMBF when first using it for the convenience of the general reader.

p.20, l.17-17: Here you first discuss the influence of the model resolution on the representativity of the station observations. This is not linked to the next sentence, which raises the question the resolution's influence on the need for scaling. This is an example for my second point made above.

Figure 9: It would help to print the scaling factor also in the graphics and you may consider merging this with Figure 3 to make the comparisons easier for the reader.

REFERENCES

W. Seiler, P. J. Crutzen.  Estimates of gross and net fluxes of carbon between the biosphere and the atmosphere from biomass burning. Climatic Change, 2(3):207–247, 1980.

---

## Author Comment (AC1) · 4 May 2016

**Response to Review**

We thank the referees for their comments on our paper. We have responded to all the referee comments through the responses below and through modifications to our manuscript. This process has improved our manuscript, which we hope is now suitable for publication. To guide the review process, referee comments are in plain text, our responses are in italics, additions to our manuscript are shown below in red and as tracked changes in the revised manuscript.

As a result of the reviewer comments regarding AOD, we have made improvements to the way in which we calculate AOD from the model output. We now calculate the AOD assuming internally mixed aerosol, which is more consistent with our modelling approach in GLOMAP. We also found that the way in which the extinction was calculated previously was not consistent with our modelling approach. Extinction efficiency from only one aerosol component was being used in the calculation of AOD rather than the extinction for each specific aerosol component. Correcting this in the code, substantially improved our simulated AOD values relative to the observations (with both the external and internal mixing assumptions). We have updated statistical values and figures in the manuscript and altered the description of the AOD calculation in the text to describe the revised method assuming internal mixing (please see the revised manuscript for details).

We have now also included additional sensitivity tests in the paper for the AOD calculation, testing the assumptions of mixing (internal versus external) and calculation of water uptake. Please see the responses to individual reviewer comments below and the revised manuscript for further detail.

**Anonymous Referee #1**

The paper of Reddington et al. investigates the impacts of biomass burning on tropical aerosols. This is done with GLOMAP global aerosol model, evaluated by long-term surface observation of PM2.5 and AOD. Specifically, this work compares three different fire emission datasets (GFED3, GFAS1 and FINN) to explore the uncertainty in emissions.

This study aims to "better understand the discrepancy in modeled biomass burning AOD and to ultimately improve estimates of biomass burning aerosol". While the authors address the contribution of underestimation in biomass burning aerosols to the bias in AOD, it would have been beneficial if they could perform further analysis to see the relative contribution of other factors. For example, if they assume internal mixing instead of external mixing, what does this do to the modeled AOD bias? Is the uncertainty in RH large enough to explain the bias in modeled AOD?

*We agree that these are important next steps. However, further isolating the reason for model bias will be very difficult without additional observations. In future work we are using detailed observations of the vertical profile of aerosol and relative humidity to better understand the causes for model bias. We hope that this future work will allow us to explore the contribution of different factors to model bias, including the uncertainty in RH as suggested by the referee.*

*As suggested, we have tested the sensitivity of internal versus external mixing in our calculation of AOD. We have included an addition figure (Fig. 7 in the revised manuscript) and added the following text to Section 4.1.3:*

"We find that the difference in AOD between assuming an external mixture of aerosol species and an internal (volumetrically-averaged) mixture is limited. Figure 7 shows the simulated versus observed multi-annual monthly mean AOD at AERONET sites when assuming external and internal mixing and indicates that the difference is less than 5%, internal mixing generally yielding higher AOD at the AERONET site locations. However, we note that the internal mixing assumption does not take into account the lensing effects of coating BC with organic aerosol, which has been shown to interact with the aerosol absorption in a non-linear way (Saleh et al., 2015)."

*We have also included an additional test to estimate the sensitivity of the simulated AOD to the calculated hygroscopic growth of the aerosol (please see Section 4.1.3 in the revised manuscript).*

In summary, this paper is well written. It describes what they did and is easy to follow along. It adds value to the literature on this topic and is worthy of publication in ACP subject to addressing these.

*We thank the referee for these positive comments.*

**Other minor things:**

1. Are the model results also obtained for the year of 2003-2011? I did not see it in the text.

*Yes, we have now clarified this in the text (P6, L24-25, revised manuscript):*

"Simulations were run for the period 2003 to 2011."

2. I found it difficult to read the tiles for x/y axis and legend in figure 4&8.

*We will ensure that the axes titles and legends are legible in the ACP version of our paper.*

**Anonymous Referee #2**

**General comments**

The paper evaluates multi-annual (2003-2011) monthly mean values of near-surface aerosol mass concentration (PM2.5) and aerosol optical depth (AOD) simulated by the GLOMAP global aerosol model against corresponding measurements conducted at four ground stations in the Amazon region and at 27 AERONET stations located in tropical regions worldwide. The simulations were done with three different datasets of biomass burning emissions (GFED3, FINN1, and GFAS1). Additional numerical experiments involved scaling the biomass burning emissions by a factor of 1.5 or 3.4. The model performance is evaluated in terms of the Pearson correlation coefficient and the normalized mean bias factor (NMBF). It is found that the model considerably underestimates both PM2.5 concentrations and AOD, with a greater underestimation of AOD than PM2.5. The paper is well written and the presentation quality is good. However, the scientific significance and the overall scientific quality of the study are questionable.

Indeed, the fact that models tend to underestimate AOD over regions affected by fires (including the Amazon region) is well known. This is acknowledged by the authors: some earlier studies reporting the underestimation of AOD are mentioned in the paper, although the list of such studies is certainly not complete (see also, e.g., Petrenko et al., 2012;

Konovalov et al., 2014). The use of ground measurements of surface aerosol mass concentration together with AOD measurements is a relatively novel point. However, the parallel analysis of PM2.5 and AOD measurements, as well as the use of three different emission datasets also did not help to fully explain the mismatch between the model and measurements of AOD. Note that Petrenko et al. (2012) has presented a much more extensive analysis of the impact of biomass burning emission.

*We agree with the referee that a novelty in our analysis is using surface PM2.5 concentrations in addition to AOD. We also agree that whilst our study does not fully explain the discrepancy between model and measured AOD, it provides additional evidence for potential reasons. In future work we aim to exploit extensive observations of aerosol properties from the SAMBBA field campaign over the Amazon to further explore this issue. We thank the reviewer for pointing us to these additional papers which are now cited in the revised manuscript.*

A serious drawback of the analysis is that it is based on an obsolete / simplistic understanding of organic aerosol processes. In particular, the authors disregard the well-established facts that organic aerosol (which is a major fraction of biomass burning aerosol) is formed by organic compounds featuring a broad distribution of volatilities (see, e.g., May et al., 2013) and that a part of them, while in the gas phase, can provide a major source of secondary organic aerosol (SOA) (see, e.g., Grieshop et al., 2009; Hennigan et al., 2011) as a result of oxidation processes. Meanwhile, these facts have direct implications for biases in biomass burning aerosol emission inventories and for the mismatch between simulations and measurements of AOD (Jathar et al., 2014; Konovalov et al., 2015; Shrivastava et al., 2015). Although it is briefly mentioned that the SOA formation in biomass burning plumes can contribute to the difference between the simulations and observations, any quantitative estimates of such a contribution are not provided. A study could benefit from simulations of biomass burning aerosol by using the volatility basis set framework and available parameterizations (e.g., Hodzic at al., 2010; Shrivastava et al., 2013, 2015). In any case, simplifications made in the model in regard to organic aerosol processes (such as an implicit assumption that organic part of biomass burning aerosol consist only of non-volatile material) and their implications for the results of this analysis had to be carefully described and discussed in light of earlier findings from relevant laboratory, field and modeling studies.

*We thank the reviewer for these comments. We agree with the referee that including a treatment of SOA formation from biomass burning may be important. To address this we have extended our discussion of SOA formation in biomass burning plumes. Implementing a volatility basis set (VBS) algorithm into global aerosol microphysics models is difficult due to the large number of additional tracers this requires, as well as large parameter uncertainty. For this reason very few global aerosol microphysics models have implemented such a complex treatment of organic aerosol. Our treatment of organic aerosol is similar to many other global aerosol microphysics models (Tsigaridis et al., 2014), making our model appropriate for exploring the ability of such models to simulate organic aerosol in regions influenced by biomass burning. We note that global models with greater complexity in their treatment of organic aerosol do not necessarily better simulate observed organic aerosol (Tsigaridis et al., 2014).*

*As suggested by the referee, we have added a discussion of how the treatment of organic aerosol may impact our results. We have added text in the introduction:*

"The contribution of secondary organic aerosol (SOA) from the oxidation of volatile organic compounds in biomass burning plumes is also a large uncertainty (Jathar et al., 2014; Shrivastava et al., 2015)."

*as well as in Section 4.1:*

"In future work we need to include the formation of semi-volatile SOA in biomass burning plumes that has been shown to be important (Konovalov et al., 2015; Shrivastava et al., 2015)."

*and the conclusions:*

"We have treated biomass burning emissions as primary and non-volatile. Formation of semi-volatile SOA in biomass burning plumes may be important (Konovalov et al., 2015; Shrivastava et al., 2015) and needs to be explored in future work."

**Specific comments**

1. Page 6. The PM2.5 measurements made by the gravimetric filter analysis method that is known to be associated with large uncertainties (Malm et al., 2011). The authors estimate uncertainties of such measurements to be 15%. But how was the loss of organic aerosol mass due to desorption estimated? Available volatility distributions of fresh biomass burning emissions (e.g., May et al., 2013) imply that the loss of the organic aerosol mass from samples taken inside biomass burning plumes (POM~1000 ug/m3) after equilibration to ambient conditions (POM~10 ug/m3) could be as large as 40 percent.

*We agree with the reviewer that the filter measurements used in our paper are associated with uncertainties and subjected to positive biases (mostly due to water) and negative biases (volatilization of semivolatile organics). Aerosols could volatilize along the filter exposure due to ambient temperature variations. Fortunately, in Amazonia the diurnal variation of temperature is relatively low (<~3ºC), which helps to limit volatilization. In addition, before gravimetric analysis, the filters are kept at controlled conditions of 20ºC and 50% RH. This temperature of 20ºC is usually below the temperature at which the samples were taken (25.2 ± 1.6ºC annual mean in Porto Velho), which also helps to prevent volatilization.*

*At the current time, we do not have means to quantify the losses due to volatilization as done by Malm et al. (2011), because the system used was simpler compared to the system used by the IMPROVE network. What we can do are mass reconstructions and also comparisons between filter-based PM2.5 and particle mass derived from other measurements, such as size distribution, AMS measurements and BC measurements. Such comparisons yielded the 15% uncertainty estimate stated in the paper.*

*Regarding the loss of POM from fresh biomass burning plumes, the filter measurements were not intended to accurately describe PM2.5 concentrations close to the biomass burning source, but instead intended to describe the variability of PM2.5 concentrations at Amazonian sites impacted by already aged biomass burning plumes (aging of 3-12 h, depending on the site and on the distribution of fire spots).*

2. Page 9, line 12. The fire emissions were injected into the model by using a set of fixed ecosystem-dependent altitudes. Meanwhile, it is known that the injection height depends on the fire intensity. If, for example, the injection height for major fires was underestimated in the model, the surface PM2.5 concentration during fire seasons could be overestimated. The study could benefit from using one of more realistic parameterizations of the injection height (e.g., Sofiev et al., 2012; Paugam et al., 2016). And, anyway, it would be important to ensure by means of a sensitivity analysis that the discrepancies between the results obtained with PM2.5 and AOD measurements are not due to biases in the injection height. The adequacy of the injection heights could further be evaluated by using surface measurements of CO concentrations and satellite observations of CO columns (see, e.g., Konovalov et al., 2014).

*This is a good suggestion and would be interesting to explore in future work. However, including a plume rise parameterization in our model would be a substantial piece of research that is not possible in our present study. We also note that using a plume rise model does not always lead to improved agreement with observations in biomass burning regions (e.g. Archer-Nicholls et al., 2015).*

*Extensive efforts to constrain fire injection heights have been described elsewhere and are not a specific focus of this work. We add the following text to Sect. 3.1:*

"Analysis of smoke plume heights has demonstrated that most smoke emissions from fires occur within the boundary layer (Val Martin et al., 2010)."

*References:*

*Archer-Nicholls, S., Lowe, D., Darbyshire, E., Morgan, W. T., Bela, M. M., Pereira, G., Trembath, J., Kaiser, J. W., Longo, K. M., Freitas, S. R., Coe, H., and McFiggans, G.: Characterising Brazilian biomass burning emissions using WRF-Chem with MOSAIC sectional aerosol, Geosci. Model Dev., 8, 549-577, doi:10.5194/gmd-8-549-2015, 2015.*

*Val Martin, M., Logan, J. A., Kahn, R. A., Leung, F.-Y. , Nelson, D. L. and Diner, D. J.: Smoke injection heights from fires in North America: Analysis of 5 years of satellite observations, Atmos. Chem. Phys., 10, 1491–1510, doi:10.5194/acp-10-1491-2010, 2010.*

3. Page 10, line 13. "The water uptake for each soluble aerosol component is calculated on-line in the model according to ZSR theory". Was hygroscopicity of organic components of biomass burning aerosol taken into account in the simulations? If so, what were typical values of the hygroscopic growth factor for the organic fraction?

*Yes, the hygroscopicity of organic components of biomass burning aerosol is taken into account in the simulations. The water content of each mode in GLOMAP given component concentrations (in air) is calculated using ZSR and binary molalities evaluated using water activity data from Jacobson (2005; Table B.10, p. 748). The particulate organic matter (POM) component is assumed to be water-insoluble in the insoluble mode but is assumed to have aged chemically in the aerosol to become hygroscopic once transferred to the soluble modes. To represent this in the ZSR calculation, the aged POM component is assumed to take up water at a fraction (set at 0.65) of sulphate.*

*We have added the following text to Section 3.2:*

"The water uptake for each soluble aerosol component is calculated on-line in the model according to Zdanovskii-Stokes-Robinson (ZSR) theory, which estimates the liquid water content as a function of solute molarity (Stokes and Robinson, 1966). We assign moderate hygroscopicity to POM in the soluble modes, consistent with a water uptake per mole at 65% of $SO_4$ (Mann et al., 2010)."

*References:*
*Jacobson, M. Z. (2005), Fundamentals of Atmospheric Modeling, 2$^{nd}$ Edn, Cambridge University Press.*
*Stokes, R. H. and Robinson, R. A.: Interactions in aqueous nonelectrolyte solutions. I. Solute-solvent equilibria, J. Phys. Chem., 70, 2126–2130, 1966.*

4. Section 3.3. The paper could significantly benefit from an analysis of inter-annual variability of fire emissions and of corresponding PM2.5/AOD values in the Amazon region during the fire season. Has such a variability been predicted by the different inventories consistently? Can the model reproduce the observed inter-annual variability in PM2.5? Which of the inventories considered does enable the best agreement between the inter-annual variations in the simulations and measurements of PM2.5?

*These are good suggestions but would add substantially to what is already a long paper. Analysis of inter-annual variability is not a specific focus of this piece of work. Also, the reason for calculating and comparing average seasonal cycles is that we were keen to ensure that the number of data points at each observation site would be equal so that the overall statistical values would not be biased by model performance at one or two locations with more years of data available.*

*In order to evaluate the model's and emission datasets' abilities to reproduce the observed inter-annual variability, we have included additional figures in the supplementary material (Figs. S2 and S3) to show the modelled versus observed annual mean PM2.5 concentrations and AOD. We have also added the following text to Sect. 4.1.1:*

"If we consider the inter-annual variability in simulated and observed PM2.5 concentrations (Figure S2), we find that the results are consistent with the evaluation of the simulated seasonal cycle. The smallest bias between model and observations is with the FINN1 emissions (NMBF= -0.22) compared to GFED3 (NMBF= -0.36) or GFAS1 (NMBF= -0.48). One notable point is that the model with GFED3 emissions simulates the highest PM2.5 concentrations for the 2010 drought year, relative to the model with GFAS1 or FINN1 emissions, leading to improved agreement with observations at Porto Velho (see Figs. 3a, 4a and S2)."

5. Page 15, line 17. "This suggests that the negative model bias in the dry season is largely due to uncertainty in the biomass burning emissions rather than anthropogenic emissions, SOA or microphysical processes in the model." Please see above a general comment about the potential importance of SOA formation in biomass burning plumes.

*We have reworded this statement to clarify we mean biogenic SOA:*

"This suggests that the negative model bias in the dry season is largely due to uncertainty in the biomass burning rather than anthropogenic emissions, biogenic SOA or microphysical processes in the model."

6. Page 20, line 20. "Uncertainties exist in the calculation of AOD that may contribute to the negative bias in simulated AOD." Did the authors try to validate their AOD calculations with other independent data? For example, it would be interesting to see if the model calculations are consistent with available measurements of the mass scattering and absorbing efficiencies (e.g. Reid et al., 2005). A bias in these parameters would indicate a similar bias in the AOD calculations.

*This is a good suggestion. However, there are a number of difficulties involved in comparing simulated values with the measurements in Reid et al. (2005). Firstly, the mass absorbing efficiencies (MAEs) obtained by Reid and Hobbs (1998) were for fires observed in the 1995 burning season; we only have GFED3, GFAS1 and FINN1 model simulations for the 2003-2011 period, where burning conditions may be quite different to those observed in 1995. Secondly the measured values are for smoke less than 4 minutes old, which a global model is unlikely to be able to capture. For these reasons we have included a comparison between the simulated and observed values rather than a detailed evaluation. We have also included a comparison between the GLOMAP simulated values with those of other models. We have added the following to the supplementary material:*

"Reid and Hobbs (1998) report values of mass absorption efficiency (MAE) for smouldering ($0.7\pm0.1$ m$^2$ g$^{-1}$) and flaming ($1.0\pm0.2$ m$^2$ g$^{-1}$) forest fires in Brazil, sampled between 13$^{th}$ August and 25$^{th}$ September 1995. To evaluate the simulated mass extinction efficiency (MEE) against observations, we calculated values of MEE from the observed MAE and single scattering albedo (SSA) from Reid and Hobbs (1998), assuming: MAE = MEE * (1-SSA). For smouldering forest fires we obtained an "observed" MEE (550 nm) of 4.4 m$^2$ g$^{-1}$ (range: 3.3 to 5.7 m$^2$ g$^{-1}$, calculated from the quoted standard errors). To compare to the observed value, we calculated MEEs at 550 nm for each simulation (with fire emissions), in grid cells that

cover the locations where smoke from the forest fires were sampled (in the vicinity of Porto Velho, Rondônia and Marabá, Pará), and calculated an average for August over the period 2003-2011.

The average simulated MEE values of 5.2-5.4 $m^2$ $g^{-1}$ (using the ZSR water uptake scheme to calculate aerosol hygroscopic growth) and 3.4-3.5 $m^2$ $g^{-1}$ (using the κ-Köhler water uptake scheme) span the observed value and are within the uncertainty range of the observations. The range in the simulated values (e.g. 5.18-5.35 $m^2$ $g^{-1}$) demonstrates the relatively limited sensitivity of the MEE to the fire emission dataset (average values are within 5%) compared to the sensitivity to the calculation of aerosol hygroscopic growth (with average values differing by a factor of 1.5). The comparison between simulated and observed MEEs supports the conclusion in the main text (Sect. 4.1.3) that the ZSR and κ-Köhler AOD are likely to represent high and low water uptake cases, respectively.

We also compare the GLOMAP simulated global mean values for aerosol burden, AOD, and MEE against those of other global aerosol models (see Table S2). In general we find that the GLOMAP global mean aerosol burdens and AOD (550 nm) are consistent with values from AEROCOM (Kinne et al., 2006) and Heald et al. (2014) for $SO_4$, BC and sea salt. For the POM and mineral dust components, both the burden and AOD are underestimated by GLOMAP relative to the other models. There could be several reasons for this underestimation (including different anthropogenic emissions and/or aerosol removal schemes in the models), but one factor that may partly explain the higher burden and AOD values for POM from the GEOS-Chem model relative to GLOMAP is the higher assumed POM:OC ratio of 2 (Heald et al., 2014), compared to 1.4 assumed in GLOMAP. The GLOMAP simulated global mean MEEs for all components are within the large range in values reported by AEROCOM (Kinne et al., 2006; Mhyre et al., 2013) and Heald et al. (2014). The MEEs for POM, $SO_4$ and BC calculated using the ZSR water uptake scheme are generally at the upper end of the AEROCOM values (particularly for BC), and those calculated using the κ-Köhler water uptake scheme are towards the lower end."

*References added:*

*Heald, C. L., Ridley, D. A., Kroll, J. H., Barrett, S. R. H., Cady-Pereira, K. E., Alvarado, M. J., and Holmes, C. D.: Contrasting the direct radiative effect and direct radiative forcing of aerosols, Atmos. Chem. Phys., 14, 5513-5527, doi:10.5194/acp-14-5513-2014, 2014.*

*Kinne, S., Schulz, M., Textor, C., Guibert, S., Balkanski, Y., Bauer, S. E., Berntsen, T., Berglen, T. F., Boucher, O., Chin, M., Collins, W., Dentener, F., Diehl, T., Easter, R., Feichter, J., Fillmore, D., Ghan, S., Ginoux, P., Gong, S., Grini, A., Hendricks, J., Herzog, M., Horowitz, L., Isaksen, I., Iversen, T., Kirkevåg, A., Kloster, S., Koch, D., Kristjansson, J. E., Krol, M., Lauer, A., Lamarque, J. F., Lesins, G., Liu, X., Lohmann, U., Montanaro, V., Myhre, G., Penner, J., Pitari, G., Reddy, S., Seland, O., Stier, P., Takemura, T., and Tie, X.: An AeroCom initial assessment – optical properties in aerosol component modules of global models, Atmos. Chem. Phys., 6, 1815–1834, doi:10.5194/acp-6-1815-2006, 2006.*

*Myhre, G., Samset, B. H., Schulz, M., Balkanski, Y., Bauer, S., Berntsen, T. K., Bian, H., Bellouin, N., Chin, M., Diehl, T., Easter, R. C., Feichter, J., Ghan, S. J., Hauglustaine, D., Iversen, T., Kinne, S., Kirkevag, A., Lamarque, J. F., Lin, G., Liu, X., Lund, M. T., Luo, G., Ma, X., van Noije, T., Penner, J. E., Rasch, P. J., Ruiz, A., Seland, O., Skeie, R. B., Stier, P., Takemura, T., Tsigaridis, K., Wang, P., Wang, Z., Xu, L., Yu, H., Yu, F., Yoon, J. H., Zhang, K., Zhang, H., and Zhou, C.: Radiative forcing of the direct aerosol effect from AeroCom Phase II simulations, Atmos. Chem. Phys., 13, 1853–1877, doi:10.5194/acp-13-1853-2013, 2013.*

*Reid, J. S. and Hobbs, P. V.: Physical and optical properties of smoke from individual biomass fires in Brazil, J. Geophys. Res., 103, 32 013–32 031, 1998.*

**Minor comments**

1. Page 11, line 17. Daily GFED3 fire emissions were implemented in GLOMAP for the period 2003–2011, with monthly emissions implemented for the period 1997–2002. Were simulations analyzed in this paper really extended to the period 1997–2002?

*We only analyse model simulations for 2003-2011, so we have removed the statement discussing simulations for 1997-2002 (which were only performed with GFED3 emissions). We have clarified the simulation period in the text (P6, L24-25, revised manuscript):*

"Simulations were run for the period 2003 to 2011."

*and (P9, L9-10, revised manuscript):*

"We complete GLOMAP simulations for the period 2003 to 2011 where all three emission datasets are available."

2. Several papers cited in the text (Chin et al., 2009; Randerson et al., 2012; Zhou et al. 2002 : : : ) are missing in the references.

*Thank you for spotting this. We have added these papers to the references and have carefully checked that all cited papers are now included in the reference list.*

**Anonymous Referee #3**

This manuscript evaluates global aerosol model simulations that have been performed with the GLOMAP model and three widely used fire emission inventories, namely GFED3, GFAS1 and FINN1. The simulations are validated thoroughly and in considerable detail with AOD and PM2.5 measurements performed in the tropics, i.e. South America, Africa and SE Asia. The study addresses the most pertinent issues recently discussed in the field of smoke aerosol modelling, i.e. the omission of small fires in burnt-area-based inventories and the need to scale up the pyrogenic aerosol flux for use in global atmospheric models. The statistical analysis is based on monthly mean values. The study is therefore very well suited as a guide on how to best select one of the fire emission inventories for use with GLOMAP, and on how accurate the simulated smoke AOD and PM2.5 may be. Considering the wide use of GLOMAP and of the investigated emission inventories, the study presents relevant results that are worth publication in ACP.

*We thank the referee for these positive comments on our manuscript.*

The study is well written and clearly presented. It adds quantitative detail to the already existing characterization of the fire emission inventories. However, this quantitative detail appears to be linked to using the GLOMAP model, and it cannot necessarily be transferred to use in other atmospheric aerosol models. The authors missed several opportunities to obtain more generally applicable new results. In particular:

- Correlations are calculated from monthly averages like so many studies have done in the past. Since emission, model and observation data are available with daily resolution, investigating this time scale would have been easily possible and much more novel.

*This is a good suggestion. However, most of the aerosol observations are not available consistently at 24-hour resolution, but are often averages over several days (see Page 6, Line 15-16, ACPD version). Thus a detailed comparison at daily time resolution is not possible with this dataset.*

*We have, however, put a lot of effort into performing a more accurate comparison between than model and observations than simply comparing monthly means. Prior to calculating the monthly averages, we removed all invalid or missing observation 'days' from the model time-series. In addition, if the PM2.5 measurement extended over a period of more than 1 day then we averaged the model data over same number of days. Therefore, the model and observed monthly means are calculated over the same days in each month. This is*

*particularly important when considering the model evaluation against AERONET AOD, since the Level 2 AERONET data contain numerous gaps in the time-series. In future work we will evaluate the model at sub-monthly time scales where we have observations consistently available at higher time resolution.*

*To give a qualitative comparison between model and observations at higher than monthly time resolution, we have included a additional figure in the supplementary material (Figure S1) showing the full time series (between 2003 and 2011) of un-averaged observed PM2.5 concentrations with daily modelled PM2.5 concentrations.*

- The study shows that PM2.5 and AOD require different upscaling of emissions. It would have been most interesting and new to study possible reasons for this. I suspect, it points to model shortcomings, but in which part of the model?
- Likewise, it would have been of general interest to see whether any of the model configuration parameters have an impact on the amount of upscaling required for any given inventory.

*We agree that these are important next steps. However, it is difficult to make progress on this issue without additional observations. In future work we are using detailed aircraft observations of the vertical profile of aerosol combined with ground and satellite AOD and the model to further explore model deficiencies and isolate the probable cause.*

*Now that we have improved the calculation of AOD, we find that the model biases in PM2.5 and AOD are more consistent in South America, although not at every location. To investigate this discrepancy further, we have performed some additional sensitivity studies with the simulated AOD. We tested the sensitivity of simulated AOD to assumptions about the aerosol mixing state and hygroscopic growth. We find that the simulated AOD is very sensitive to the calculation of water uptake, which could have a large impact on the amount of upscaling of emissions required for the model match observed AOD. This highlights how the use of an emissions scaling factor could be compensating for inadequate understanding of water uptake by the aerosol and the subsequent changes in aerosol size distribution and optics.*

I am aware that addressing one of these issues in the final manuscript will imply a major effort, which may not be justified at this stage. However, if the authors would be willing to do it, this would certainly make the results applicable for a much larger community, i.e. also those who use other models than GLOMAP or its results.

Since the manuscript is very well written, I have only very few minor comments:

**SPECIFIC COMMENTS**

1. p.11, l.1 and p.12, l.13: delete "yearly varying"

*Deleted as suggested.*

2. p.11, l.28: You may cite Seiler Crutzen 1980 for this formula.

*Done.*

3. p.14, l.16: Please add the definition of NMBF when first using it for the convenience of the general reader.

*We have added the following to Sect.4.1.1:*

"To quantify the agreement between model and observations, we use the Pearson correlation coefficient (r) and normalised mean bias factor (NMBF) as defined by Yu et al. (2006):

$$NMBF = \frac{(\sum M_i - \sum O_i)}{|\sum M_i - \sum O_i|}\left[\exp\left(\left|\ln\frac{\sum M_i}{\sum O_i}\right|\right) - 1\right]$$

where *M* and *O* represent the multi-annual monthly mean model and observed values, respectively, for each month *i*. A positive NMBF indicates the model overestimates the observations by a factor of NMBF+1. A negative NMBF indicates the model underestimates the observations by a factor of 1–NMBF."

    4. p.20, l.17-17: Here you first discuss the influence of the model resolution on the representativity of the station observations. This is not linked to the next sentence, which raises the question the resolution's influence on the need for scaling. This is an example for my second point made above.

*We have assumed this comment refers to P20, L19-24 (ACPD version). This is a good point and we agree that the two issues have been conflated in this paragraph, although we do believe the two issues are linked. We acknowledge that a relatively coarse model resolution presents a limitation in the comparison with point measurements. However, we do interpolate the model values to the specific site locations so we have altered the paragraph in question to focus more on the potential for model resolution to decrease agreement at the sites (rather than how well the site location represents the surrounding area):*

"Another important factor that will also influence the calculated AOD is the spatial resolution of the simulated aerosol and RH (used to calculate aerosol water uptake) fields. These fields are on a relatively coarse spatial resolution and will not capture small scale (sub-grid) variability in these quantities that may influence point location measurements from AERONET stations. A higher resolution model would be required to test how sensitive the simulated AOD is to the spatial resolution of the aerosol and RH fields and whether or not increasing the resolution improves the agreement with observed AOD (and reduces the discrepancy between the model performance in AOD and PM2.5). Bian et al. (2009) showed that increasing the resolution of the RH field from 2°x2.5° to 1°x1.25° can increase the simulated AOD by ~10% in biomass burning regions (improving agreement with observations), which may partly explain the larger discrepancies in AOD than PM2.5."

*Reference added: Bian, H., Chin, M., Rodriguez, J. M., Yu, H., Penner, J. E., and Strahan, S.: Sensitivity of aerosol optical thickness and aerosol direct radiative effect to relative humidity, Atmos. Chem. Phys., 9, 2375-2386, doi:10.5194/acp-9-2375-2009, 2009.*

    5. Figure 9: It would help to print the scaling factor also in the graphics and you may consider merging this with Figure 3 to make the comparisons easier for the reader.

*Thank you for this suggestion. Figures 3 and 9 and figures 6 and 10 have now been merged and labels added.*

**REFERENCES**
W. Seiler, P. J. Crutzen. Estimates of gross and net fluxes of carbon between the biosphere and the atmosphere from biomass burning. Climatic Change, 2(3):207–247, 1980.

---

## Author Response (AR2)

**Response to editor comments**

**Reddington et al.**

I find that the reviewers' concerns have been addressed adequately and that the manuscript is almost ready to be published. In some instances where the reviewers' comments would have required significant additional effort, e.g., Reviewer 2's request to address the issue of additional SOA formation from the BB volatiles, you chose to state that this is outside of the scope of the present paper. While this may be a missed opportunity to increase the scientific impact of your paper, I will accept your decision not to expand the scope of your work.

We thank the editor for this decision. We have responded to the comments below and have added discussion to our manuscript as suggested. New text in the manuscript is highlighted in yellow.

Reading the manuscript made me wonder if there might not be some compensating errors that explain some of the observed biases. For example, I noticed that you chose a very low OC-to-POM conversion factor (1.4). I think that for aged BB aerosols a factor of 2 would be more appropriate. On the other hand, sulfate in ZSR is assumed to behave like sulfuric acid, which is very unrealistic. In BB emissions, there is always enough ammonia to neutralize sulfuric acid, and ammonium to sulfate molar rations are between 1 and 2. You are also using a fairly low kappa-org (0.07). Our extensive measurements in the Amazon consistently give a value of 0.10. This then leads to the question: Would a kappa-Koehler approach with more realistic values of OC/POM and kappa-org result in a significantly reduced bias? If you don't want to change the paper, I would appreciate a response in the form of a comment.

These are good points. We have now added discussion on our OC-to-POM conversion factor and hygroscopic growth. We have added a comment pointing out that there is scope for compensating errors, particularly in calculation of AOD.

As suggested we now use a kappa value of 0.1 as observed in the Amazon (Gunthe et al., 2009). We also show that simulated AOD is sensitive to our assumptions about kappa and have added a Figure (new Figure 7) to highlight this.

We now emphasise that analysis against detailed aerosol chemical, microphysical and optical properties (Brito et al., 2014; Andreae et al., 2015) is now required to further understand any potential underestimates in biomass burning emissions.

A few minor and mostly technical issues still need to be addressed:

Thanks for pointing out these issues, which we have now addressed.

The labels and legends in several figures are much too small to readable in print. Please provide figures with larger font sizes. Reviewer 1 had requested this already for Figs. 4 and 7(old), but you did not make any changes in the figures he mentioned. The problem also extends to other figures, e.g., Fig. 1.

We have updated our figures with bigger labels and legends where possible. We will ensure that labels and legends are readable in the final typeset version of the paper.

Manuscript pages should be numbered.

Done.

According to SI recommendations, "year" in units should be abbreviated "a", not "yr".

Done.

**References**

Andreae, M. O., Acevedo, O. C., Araùjo, A., Artaxo, P., Barbosa, C. G. G., Barbosa, H. M. J., Brito, J., Carbone, S., Chi, X., Cintra, B. B. L., da Silva, N. F., Dias, N. L., Dias-Júnior, C. Q., Ditas, F., Ditz, R., Godoi, A. F. L., Godoi, R. H. M., Heimann, M., Hoffmann, T., Kesselmeier, J., Könemann, T., Krüger, M. L., Lavric, J. V., Manzi, A. O., Lopes, A. P., Martins, D. L., Mikhailov, E. F., Moran-Zuloaga, D., Nelson, B. W., Nölscher, A. C., Santos Nogueira, D., Piedade, M. T. F., Pöhlker, C., Pöschl, U., Quesada, C. A., Rizzo, L. V., Ro, C.-U., Ruckteschler, N., Sá, L. D. A., de Oliveira Sá, M., Sales, C. B., dos Santos, R. M. N., Saturno, J., Schöngart, J., Sörgel, M., de Souza, C. M., de Souza, R. A. F., Su, H., Targhetta, N., Tóta, J., Trebs, I., Trumbore, S., van Eijck, A., Walter, D., Wang, Z., Weber, B., Williams, J., Winderlich, J., Wittmann, F., Wolff, S., and Yáñez-Serrano, A. M.: The Amazon Tall Tower Observatory (ATTO): overview of pilot measurements on ecosystem ecology, meteorology, trace gases, and aerosols, Atmos. Chem. Phys., 15, 10723-10776, doi:10.5194/acp-15-10723-2015, 2015.

Brito, J., Rizzo, L. V., Morgan, W. T., Coe, H., Johnson, B., Haywood, J., Longo, K., Freitas, S., Andreae, M. O., and Artaxo, P.: Ground-based aerosol characterization during the South American Biomass Burning Analysis (SAMBBA) field experiment, Atmos. Chem. Phys., 14, 12069-12083, doi:10.5194/acp-14-12069-2014, 2014.

Gunthe, S. S., King, S. M., Rose, D., Chen, Q., Roldin, P., Farmer, D. K., Jimenez, J. L., Artaxo, P., Andreae, M. O., Martin, S. T., and Pöschl, U.: Cloud condensation nuclei in pristine tropical rainforest air of Amazonia: size-resolved measurements and modeling of atmospheric aerosol composition and CCN activity, Atmos. Chem. Phys., 9, 7551-7575, doi:10.5194/acp-9-7551-2009, 2009.